# Stochastic social behavior coupled to COVID-19 dynamics leads to waves, plateaus, and an endemic state

**Alexei V Tkachenko[1]\*, Sergei Maslov[2]\*, Tong Wang[3], Ahmed Elbana[4], George N Wong[5], Nigel Goldenfeld[6]**

[1]Center for Functional Nanomaterials, Brookhaven National Laboratory, Upton, United States; [2]Department of Bioengineering, University of Illinois Urbana-Champaign, Urbana, United States; [3]Department of Physics, University of Illinois at Urbana-Champaign, Urbana, United States; [4]Department of Civil Engineering, University of Illinois at Urbana-Champaign, Urbana, United States; [5]Carl R. Woese Institute for Genomic Biology, University of Illinois at Urbana-Champaign, Urbana, United States; [6]University of Illinois at Urbana-Champaign, Urbana, United States

**Abstract** It is well recognized that population heterogeneity plays an important role in the spread of epidemics. While individual variations in social activity are often assumed to be persistent, that is, constant in time, here we discuss the consequences of dynamic heterogeneity. By integrating the stochastic dynamics of social activity into traditional epidemiological models, we demonstrate the emergence of a new long timescale governing the epidemic, in broad agreement with empirical data. Our stochastic social activity model captures multiple features of real-life epidemics such as COVID-19, including prolonged plateaus and multiple waves, which are transiently suppressed due to the dynamic nature of social activity. The existence of a long timescale due to the interplay between epidemic and social dynamics provides a unifying picture of how a fast-paced epidemic typically will transition to an endemic state.

**\*For correspondence:**
oleksiyt@bnl.gov (AVT);
maslov@illinois.edu (SM)

**Competing interest:** The authors declare that no competing interests exist.

## Editor's evaluation

This is an excellent and elegant example of what theory can do at its best in epidemiology: it takes a widely observed phenomenon that is an 'embarrassment' (my word) to current theories; proposes a parsimonious explanation that is plausible for the phenomenon by extending the existing theories in a specific way; and makes a plausible case for the importance of the mechanism in explaining key features of the data. In this case, the embarrassing phenomenon is long periods of very slowly changing incidence/prevalence, and the modification to theory is incorporation of dynamic social heterogeneity. This should stimulate much further work in the field. Congratulations to the authors.

## Introduction

The COVID-19 pandemic has underscored the prominent role played by population heterogeneity in epidemics. It has been well documented that at short timescales the transmission of the infection is highly heterogeneous. That is to say, it is characterized by the phenomenon of superspreading, in which a small fraction of individuals is responsible for a disproportionately large number of secondary infections (*Lloyd-Smith et al., 2005*; *Galvani and May, 2005*; *Endo et al., 2020*; *Sun et al., 2021*). At the same time, according to multiple models, persistent population heterogeneity is expected to suppress the herd immunity threshold (HIT) and reduce the final size of an epidemic (*Pastor-Satorras*

*et al., 2015*; *Bansal et al., 2007*; *Gomes et al., 2020*; *Tkachenko et al., 2021*; *Neipel et al., 2020*; *Britton et al., 2020*). In the context of COVID-19, this observation led to a controversial suggestion that a strategy relying exclusively on quickly reaching herd immunity might be a viable alternative to government-imposed mitigation. However, even locations that have been hardest hit by the first wave of the epidemic have not gained a lasting protection against future waves (*Faria et al., 2021*; *Sabino et al., 2021*). Another puzzling aspect of the COVID-19 pandemic is the frequent occurrence of plateau-like dynamics, characterized by approximately constant incidence rate over a prolonged time (*Thurner et al., 2020*; *Weitz et al., 2020*).

These departures from predictions of both classical epidemiological models and their heterogeneous extensions have led to a greater appreciation of the role played by human behavior in epidemic dynamics. In particular, one plausible mechanism that might be responsible for both suppression of the early waves and plateau-like dynamics is that individuals modify their behavior based on information about the current epidemiological situation (*Epstein et al., 2008*; *Funk et al., 2009*; *Fenichel et al., 2011*; *Bauch, 2013*; *Rizzo et al., 2014*; *Weitz et al., 2020*; *Arthur et al., 2021*). Another possibility is that long plateaus might arise because of the underlying structure of social networks (*Thurner et al., 2020*).

Here, we study epidemic dynamics, accounting for random changes in levels of individual social activity. We demonstrate that this type of dynamic heterogeneity, even without knowledge-based adaptation of human behavior (e.g., in response to epidemic-related news) (*Epstein et al., 2008*; *Funk et al., 2009*; *Fenichel et al., 2011*; *Bauch, 2013*; *Rizzo et al., 2014*; *Weitz et al., 2020*; *Arthur et al., 2021*), leads to a substantial revision of the epidemic progression, consistent with empirical data for the COVID-19 pandemic. In a recent study (*Tkachenko et al., 2021*), we have pointed out that population heterogeneity is a dynamic property that roams across multiple timescales. A strong short-term overdispersion of the individual infectivity manifests itself in the statistics of superspreading events. At the other end of the spectrum is a much weaker persistent heterogeneity operating on very long timescales. In particular, it is this long-term heterogeneity that leads to a reduction of the HIT compared to that predicted by classical homogeneous models (*Gomes et al., 2020*; *Tkachenko et al., 2021*; *Neipel et al., 2020*; *Rose et al., 2021*; *Britton et al., 2020*). In particular, in our previous work (*Tkachenko et al., 2021*), it was demonstrated that the entire effect of persistent heterogeneity can be well characterized by a single parameter, which we call the *immunity factor* $\lambda$. This quantity is related to the statistical properties of heterogeneous susceptibility across the population and to its correlation with individual infectivity. For the important case of gamma-distributed individual susceptibilities, we show that the classical proportionality between the fraction of susceptible population $S$ and the effective reproduction number, $R_e = R_0 S$, transforms into a power-law scaling relationship $R_e = R_0 S^\lambda$. This leads to a modified version of the result for the HIT, $1 - S_{HI} = 1 - R_0^{-1/\lambda}$. However, that result assumes *persistent or time-independent heterogeneity*. In reality, the epidemic dynamics is likely to be sensitive to what happens at intermediate timescales, where the social activity of each individual crosses over from its bursty short-term behavior to a smooth long-term average. Due to this type of dynamic heterogeneity, the suppression of early waves of the COVID-19 epidemic, even without active mitigation, does not signal achievement of long-term herd immunity. Instead, as argued in *Tkachenko et al., 2021*, this suppression is associated with transient collective immunity (TCI), a fragile state that degrades over time as individuals change their social activity patterns. In this work, we present a stochastic social activity (SSA) model explicitly incorporating time-dependent heterogeneity and demonstrate that the first wave is generally followed either by secondary waves or by long plateaus characterized by a nearly constant incidence rate. In the context of COVID-19, both long plateaus and multiwave epidemic dynamics have been commonly observed. According to our analysis, the number of daily infections during the plateau regime, as well as the individual wave trajectories, are robust properties of the epidemic and depend on the current level of mitigation, degree of heterogeneity, and temporal correlations of individual social activity.

Our work implies that, once plateau-like dynamics is established, the epidemic gradually evolves towards the long-term HIT determined by persistent population heterogeneity. However, reaching that state may stretch over a surprisingly long time, from months to years. On these long timescales, both waning of individual biological immunity and mutations of the pathogen become valid concerns and would ultimately result in a permanent endemic state of the infection. Such endemic behavior is a well-known property of most classical epidemiological models (*Keeling and Rohani, 2011*). However,

the emergence of the endemic state for a newly introduced pathogen is far from being completely understood (*Wolfe et al., 2007*; *Engering et al., 2019*; *Pastor-Satorras and Vespignani, 2001a*). Indeed, most epidemiological models would typically predict complete extinction of a pathogen following the first wave of the epidemic, well before the pool of susceptible population would be replenished. A commonly accepted, though mostly qualitative, explanation for the onset of endemic behavior of such diseases as measles, seasonal cold, etc., involves geographic heterogeneity: the pathogen may survive in other geographic locations until returning to a hard-hit area with a depleted susceptible pool (*Wolfe et al., 2007*; *Engering et al., 2019*). In contrast, our theory provides a simple and general mechanism that prevents an overshoot of the epidemic dynamics and thus naturally and generically leads to the endemic fixed point.

The importance of temporal effects has long been recognized in the context of network-based epidemiological models (*Starnini et al., 2017*; *Volz and Meyers, 2007*; *Bansal et al., 2010*; *Read et al., 2008*). On the one hand, available high-resolution data on real-world temporal contact networks allow direct modeling of epidemic spread on those networks. On the other hand, building upon successes of epidemic models on static unweighted networks (*Lloyd and May, 2001*; *May and Lloyd, 2001*; *Moreno et al., 2002*; *Pastor-Satorras et al., 2015*), a variety of temporal generalizations have been proposed. These typically involve particular rules for discrete or continuous network rewiring (*Volz and Meyers, 2007*; *Bansal et al., 2010*; *Read et al., 2008*) such as in activity-based network

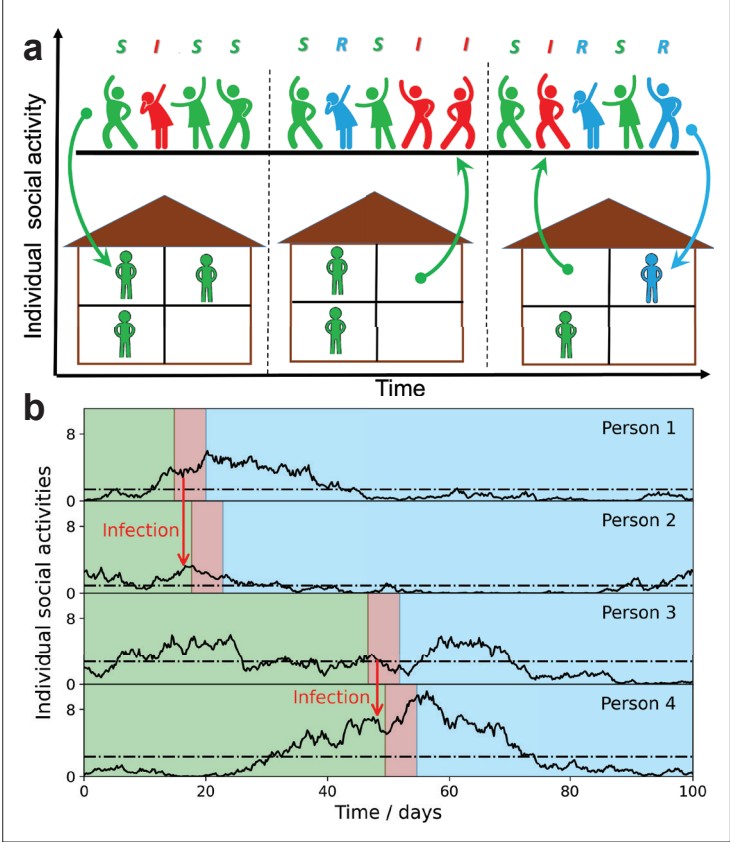

**Figure 1.** Schematic illustration of the stochastic social activity model in which each individual is characterized by a time-dependent social activity. (**a**) People with low social activity (depicted as socially isolated figures at home) occasionally increase their level of activity (depicted as a party). The average activity in the population remains the same, but individuals constantly change their activity levels from low to high (arrows pointing up) and back (arrows pointing down). Individuals are colored according to their state in the susceptible-infected-removed (SIR) epidemiological model: susceptible, green; infected, red; and removed, - blue. The epidemic is fueled by constant replenishment of susceptible population with high activity due to transitions from the low-activity state. (**b**) Examples of individual time-dependent activity $a_i(t)$ (solid lines), with different persistent levels (dot-dashed lines). S,I,R states of an individual have the same color code as in (**a**). Note that pathogen transmission occurs predominantly between individuals with high current activity levels.

models (*Perra et al., 2012*; *Vazquez et al., 2007*; *Rizzo et al., 2014*). While important theoretical results have been obtained for some of these problems, especially regarding the epidemic threshold, many open questions and challenges remain in the field. In this paper, we start with a more traditional heterogeneous well-mixed model, which is essentially equivalent to the mean-field description of an epidemic on a network (*Moreno et al., 2002*; *Pastor-Satorras and Vespignani, 2001b*; *Bansal et al., 2007*), and include effects of time-variable social activity that modulates levels of individual susceptibilities and infectivities.

## Results

### SSA model

The basic idea behind our model is represented in *Figure 1*. Each individual   is characterized by time-dependent social activity $a_i(t)$ proportional to his/her current frequency and intensity of close social contacts. This quantity determines both the individual susceptibility to infection as well as the ability to infect others. The time evolution of contact frequency, and hence $a_i(t)$, is in principle measurable by means of proximity devices, such as RFID, Bluetooth, Wi-Fi, etc. (*Salathe et al., 2010*; *Starnini et al., 2017*; *Isella et al., 2011*; *Pastor-Satorras et al., 2015*). In fact, multiple studies of that kind have been conducted over the years, alongside more traditional approaches based on, for example, personal logs (*Danon et al., 2013*). In addition, virtual interactions by means of e-mail, social media, and mobile communications are commonly used as proxies for studies of interpersonal contacts (*Rybski et al., 2009*; *Barabási, 2005*; *Saramäki and Moro, 2015*; *Nielsen et al., 2021*). Digital communications can be studied over a substantial time interval for a large number of individuals, thus presenting a significant challenge for field studies of face-to-face contact networks. It is generally accepted that the presence of an underlying dynamic contact network may drastically affect epidemic dynamics. However, the sheer complexity of that network makes it hard to integrate the social dynamics into common epidemic models. The simple stochastic model of social activity proposed in this work is based on several observations that appear to be rather generic both for real and virtual interpersonal communications. Individual social activity $a(t)$ tends to be 'bursty' and overdispersed when observed over short enough timescales (e.g., several days). While individuals demonstrate bursts of activity across multiple timescales, the analysis of various communication networks reveals a cutoff time, beyond which the level of activity reverts to its long-term average (*Vazquez et al., 2007*; *Karsai et al., 2012*). Note that this average may still exhibit person-to-person variations corresponding to persistent heterogeneity of the population. The mean-reversion time constant may range from days to months, depending on the context of the study (*Vazquez et al., 2007*; *Karsai et al., 2012*). In this work, we make a model assumption that a similar mean-reversion time $\tau_s$ exists for in-person social activity, that is, for $a_i(t)$.

In our SSA model, we combine a simple mathematical description of social dynamics with the standard susceptible-infected-removed (SIR) epidemiological model. Qualitatively it leads to long-term epidemic dynamics fueled by replenishment of the susceptible population due to changes in the level of individual social activity from low to high. *Figure 1a* illustrates this process by showing people with low social activity (depicted as socially isolated at home) occasionally increasing their level of activity (depicted as a party). *Figure 1* represents the same dynamics in terms of individual functions $a_i(t)$. Note that each person is characterized by his/her own long-term average activity level $\bar{\alpha}_i$ (dot-dashed lines), but the transmission occurs predominantly between individuals with high levels of *current* social activity. This is because $a_i(t)$ determines both the current susceptibility and the individual infectivity of a person. However, secondary transmission is delayed with respect to the moment of infection, by a time of the order of a single generation interval $1/\gamma$ (around 5 days for COVID-19).

For any individual  , the value of $a_i(t)$ has a tendency to gradually drift towards its persistent average level $\bar{\alpha}_i$, which itself varies within the population. In our model, we assign a single timescale $\tau_s$ to this mean reversion process. This is of course a simplification of the multiscale relaxation observed in real social dynamics. While $\tau_s$ can be treated as a fitting parameter of our model, here we simply set it to be $\tau_s = 30$ days, several times longer than the mean generation interval of COVID-19, $1/\gamma = 5$ days. Note that from the point of view of the epidemic dynamics, variations in activity on timescales shorter than the mean generation interval may be safely ignored. For example, attending a single party would increase an individual's risk of infection but would not change his/her likelihood of transmission to others 5 days later.

Individual social activity $a_i(t)$ is assumed to be governed by the following stochastic equation:

$$\dot{a}_i(t) = \frac{\bar{\alpha}_i - a_i(t)}{\tau_s} + \eta_i(t) \qquad (1)$$

Here, $\eta(t)$ is a zero mean Gaussian noise giving rise to time-dependent variations in $a_i(t)$. We set the correlation function of the noise as $\langle \eta_i(t)\eta_i(t') \rangle = \frac{2a_i(t)}{\tau_s k_0}\delta(t - t')$, which results in diffusion in the space of individual social activity with a diffusion coefficient proportional to $a_i$ and the correlation time $\tau_s$. This stochastic process is well known in mathematical finance as the Cox–Ingersoll–Ross (CIR) model (**Cox et al., 1985**) and has been studied in probability theory since the 1950s (**Feller, 1951**). The major properties of this model are (i) reversion to the mean and (ii) non-negativity of $a_i$ at all times, both of which are natural for social activity. Furthermore, the steady-state solution of this model is characterized by gamma-distributed $a_i$. This is consistent with the empirical statistics of short-term overdispersion of disease transmission manifesting itself in superspreading events (**Lloyd-Smith et al., 2005**; **Endo et al., 2020**; **Sun et al., 2021**). More specifically, for a given level of persistent activity $\bar{\alpha}_i$, this model generates a steady-state distribution of 'instantaneous' values of social activity $a$ following a gamma distribution with mean $\bar{\alpha}$ and variance $\bar{\alpha}/k_0$. Additional discussion of this model is presented in Appendix 1.

The statistics of superspreader events is usually represented as a negative binomial distribution, derived from a gamma-distributed individual reproduction number (**Lloyd-Smith et al., 2005**). The observed overdispersion parameter $k \approx 0.1 - 0.3$ (**Endo et al., 2020**; **Sun et al., 2021**) can be used for partial calibration of our model. This short-term overdispersion has both stochastic and persistent contributions. In our model, the former is characterized by dispersion $k_0$. In addition, we assume persistent levels of social activity $\bar{\alpha}_i$ to also follow a gamma distribution with another dispersion parameter, $\kappa$. In several recent studies of epidemic dynamics in populations with persistent heterogeneity (**Tkachenko et al., 2021**; **Aguas et al., 2020**; **Neipel et al., 2020**), it has been demonstrated that $\kappa$ determines the HIT. Multiple studies of real-world contact networks (summarized, e.g., in **Bansal et al., 2007**) report an approximately exponential distribution of $\bar{\alpha}_i$, which corresponds to $\kappa \simeq 1$. Throughout this paper, we assume a more conservative value, $\kappa = 2$, that is, coefficient of variation $1/\kappa = 0.5$, half way between the fully homogeneous case and that with exponentially distributed $\bar{\alpha}$. For consistency with the reported value of the short-term overdispersion parameter (**Sun et al., 2021**), $1/k \approx 1/\kappa + 1/k_0 \approx 3$, we set $k_0 = 0.4$.

## Epidemic dynamics with stochastic social activity

According to **Equation 1**, individuals, each with their own persistent level of social activity $\bar{\alpha}$ effectively diffuse in the space of their current social activity $a$. This leads to major modifications of the epidemic dynamics (see Appendix 1 for the detailed technical discussion). For instance, the equation for the susceptible fraction in classical epidemic models (**Keeling and Rohani, 2011**) acquires the following form:

$$\dot{S}_{\bar{\alpha}}(a, t) = \left[ -aJ(t) + \frac{a}{k_0 \tau_s}\frac{\partial^2}{\partial a^2} + \frac{\bar{\alpha} - a}{\tau_s}\frac{\partial}{\partial a} \right] S_{\bar{\alpha}}(a, t) \qquad (2)$$

Here, $S_{\bar{\alpha}}(a, t)$ is the fraction of susceptible individuals within a subpopulation with a given value of persistent social activity $\bar{\alpha}$ and with current social activity $a$, at the moment of infection, and $J(t)$ is the current strength of infection. Its time evolution can be described by any traditional epidemiological model, such as age-of-infection, SIR/SEIR, etc. (**Keeling and Rohani, 2011**).

**Equation 2** is dramatically simplified by writing it as $S_{\bar{\alpha}}(a, t) \equiv e^{-Z(t)\bar{\alpha} - k_0 h(t)a}$. The new variables $Z(t)$ and $h(t)$ measure persistent and, respectively, transient heterogeneity of the attack rate. As the epidemic progresses, new infections selectively remove people with high *current* levels of social activity $a(t)$. The variable $h(t)$ measures the degree of such selective depletion of susceptibles. Conversely, the variable $Z(t)$ quantifies the extent of depletion of susceptibles among subpopulations with different levels of *persistent* social activity $\bar{\alpha}$. In the long run, transient heterogeneity disappears due to stochastic changes in the levels of current social activity $a(t)$. Thus, $h(t)$ asymptotically approaches 0 as $t \to \infty$. We combine this ansatz with a general methodology (**Tkachenko et al., 2021**) that provides a quasi-homogeneous description for a wide variety of heterogeneous epidemiological models. For a specific case of SIR dynamics, we assign each person a state variable $I_i$ set to 1 when the individual is

infectious and 0 otherwise. Now, the activity-weighted fraction of the infected population is defined as $I(t) = \langle I_i a_i(t) \rangle_i / \langle a_i^2 \rangle_i$, and the current infection strength is proportional to it:

$$J(t) = \gamma R_0 M(t) I(t) \tag{3}$$

Here, $M(t)$ is a time-dependent mitigation factor, which combines the effects of government interventions, societal response to the epidemic, as well as other sources of time modulation, such as seasonal forcing.

Using the above ansatz, the epidemic in a population with both persistent and dynamic heterogeneity of individual social activity can be compactly described as a dynamical system with only three variables: the susceptible population fraction $S(t)$, the infected population fraction $I(t)$ (activity-weighted) that, according to *Equation 3*, is proportional to the strength of infection $J(t)$, and the transient heterogeneity variable $h(t)$. As shown in Appendix 2, the dynamics in the $(S, I, h)$ -space are given by the following set of differential equations:

$$\frac{dI}{dt} = \frac{JS^\lambda}{(1+h)^2} - \gamma I \tag{4}$$

$$\frac{dS}{dt} = -\frac{JS^{1+1/\kappa}}{(1+h)} \tag{5}$$

$$\frac{dh}{dt} = \frac{J}{k_0} - \frac{h(1+h)}{\tau_s} \tag{6}$$

As discussed above, the scaling exponent $\lambda$ in *Equation 4* is the immunity factor that we introduced in *Tkachenko et al., 2021* to describe the reduction of the HIT due to persistent heterogeneity. In the context of the present study, $\lambda$ depends both on short-term and persistent dispersion parameters as

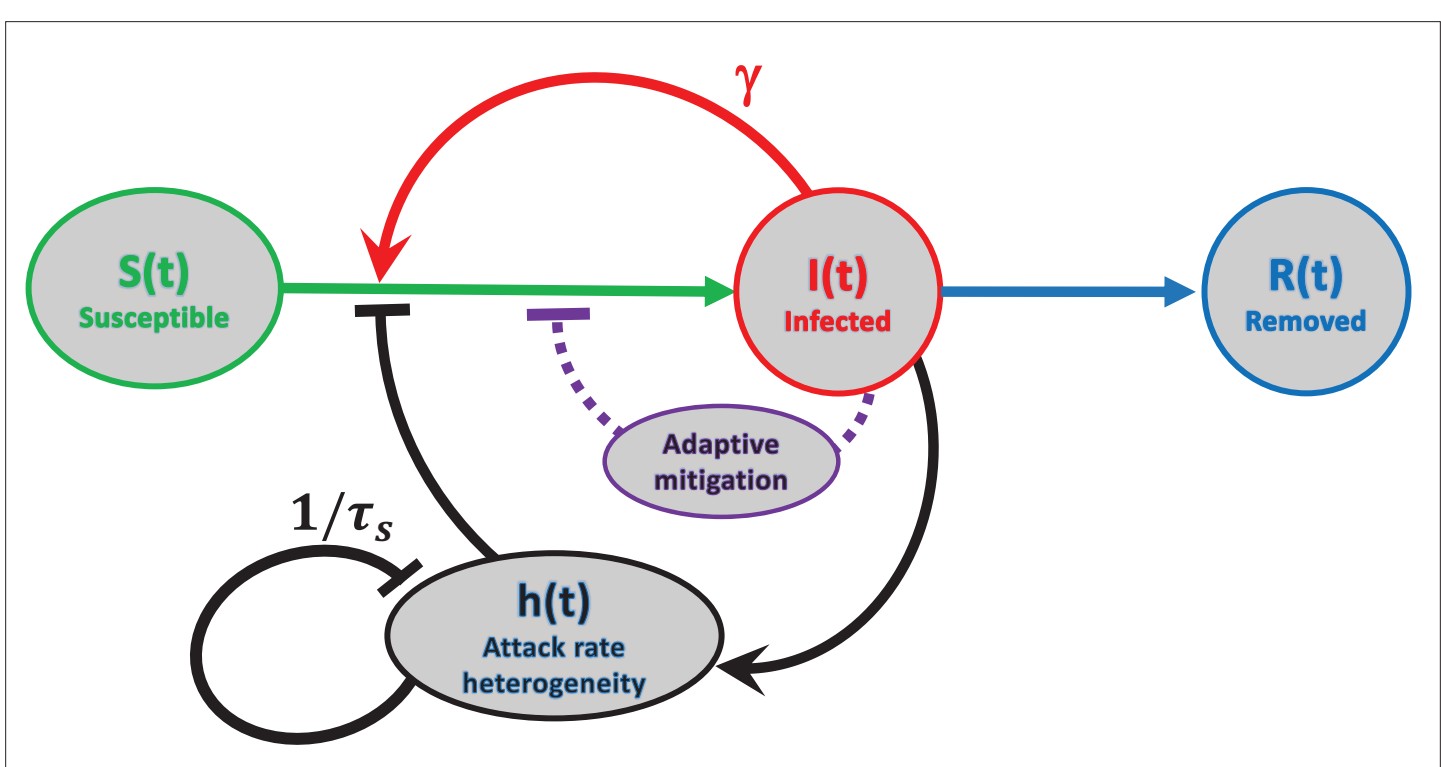

**Figure 2.** Schematic representation of feedback mechanisms that lead to self-limited epidemic dynamics. In traditional epidemic models, the major factor is the depletion of the susceptible population (red). Government-imposed mitigation and/or behavioral knowledge-based adaptation to the perceived risk creates a second feedback loop (purple). Yet another feedback mechanism is due to dynamic heterogeneity of the attack rate parameterized by $h(t)$ (black). Note that this mechanism is due to the selective removal of susceptibles with high current levels of social activity in the course of the epidemic. Therefore, it does not involve any knowledge-based adaptation, defined as modulation of average social activity in response to the perceived danger of the current level of infection. The attack rate heterogeneity $h(t)$ is generated by the current infection $J(t)$ and suppresses itself on the timescale of $\tau_s$ due to reversion of individual social activity towards the mean.

described in Appendix 2. For parameters $k_0 = 0.4$, $\kappa = 2$, and $\tau_s = 30$ days used throughout our study, one gets $\lambda = 1.7$, consistent with our earlier estimate in *Tkachenko et al., 2021*.

In *Figure 2*, we schematically represent three feedback mechanisms that lead to self-limited epidemic dynamics. The most conventional of them relies on depletion of the susceptible population (red). Another mechanism is due to government mitigation as well as personal behavioral response to perceived epidemic risk (purple). Finally, according to our theory, there is yet another generic mechanism related to accumulated heterogeneity of the attack rate, quantified by the variable $h(t)$. Due to the long-term relaxation of $h(t)$, this feedback loop limits the scale of a single epidemic wave, but does not provide long-term protection against new ones.

## Origin of waves and plateaus

As demonstrated below, the theory described by *Equation 4*, *Equation 5*, *Equation 6* is in excellent agreement with simulations of an agent-based model (ABM) in which social activities of 1 million agents undergo stochastic evolution described by *Equation 1* (compare solid lines with shaded areas in *Figure 3* and *Figure 4*).

*Figure 3* illustrates the dramatic effect that time-dependent heterogeneity has on epidemic dynamics. It compares three cases: the classical homogeneous SIR model (black), the same model with persistent heterogeneity (brown), and the dynamic heterogeneity case considered in this study (green). The latter two models share the same HIT (green dashed line) that is reduced compared to the homogeneous case (black dashed line). In the absence of dynamic heterogeneity (black and brown), the initial exponential growth halts once the respective HIT is reached, but the overall attack rate 'overshoots' beyond that point, eventually reaching a significantly larger level, known as the final size of the epidemic (FSE). Importantly, in both these cases the epidemic has only a single wave of duration set by the mean generation interval $1/\gamma$ multiplied by a certain $R_0$-dependent factor. In the case of dynamic heterogeneity (green), described by *Equation 4*; *Equation 5*; *Equation 6*, the epidemic is transiently suppressed at a level that is below even the heterogeneous HIT. As we argued in *Tkachenko et al., 2021*, this temporary suppression is due to the population reaching a state we termed transient collective immunity (TCI). That state originates due to the short-term population heterogeneity being enhanced compared to its persistent level. Stochastic contributions to social activity responsible for this enhancement eventually average out, leading to a slow degradation of the TCI state. *Figure 3b* illustrates that as the TCI state degrades, the daily incidence rate develops an extended plateau on the green curve. The cumulative attack rate shown in *Figure 3c* relaxes towards the HIT. As shown in Appendix 3, in this regime $J \sim dh/dt$. By substituting this relationship into *Equation 6*, one observes that the relaxation is characterized by an emergent long timescale. This timescale of the order of $\tau_s/k_0$ governs the relaxation towards either herd immunity or the endemic state of the pathogen. Note that it may be considerably longer than the timescale $\tau_s$ for individuals to revert to their mean level of activity provided that the short-term overdispersion is strong (i.e., $k_0 \ll 1$).

According to (*Equation 4*; *Equation 5*; *Equation 6*) for a fixed mitigation level $M(t)$, any epidemic trajectory would eventually converge to the same curve, that is, the universal attractor. The existence of the universal attractor is apparent in *Figure 4*, where we compare two scenarios with different mitigation strategies applied at early stages of the epidemic. In both cases, an enhanced mitigation was imposed, leading to a reduction of $M(t)R_0$ by 50% from 2 to 1. In the first scenario (blue curves), the enhanced mitigation was imposed on day 27 and lasted for 15 days. In the second scenario (red curves), the mitigation was applied on day 37 and lasted for 45 days. As predicted, this difference in mitigation has not had any significant effect on the epidemic in the long run: these two trajectories eventually converged towards the universal attractor. However, short- and medium-term effects were substantial. The early mitigation scenario (blue curve) resulted in a substantial suppression of the maximum incidence during the first wave. Immediately following the release of the mitigation the second wave started and reached approximately the same peak value as the first one. If the objective of the intervention is to avoid overflow of the healthcare system, this strategy would indeed help to achieve it. In contrast, the delayed mitigation scenario (red curve) turned out to be largely counterproductive. It did not suppress the peak of the first wave, but brought the infection to a very low level after it. Eventually, that suppression backfired as the TCI state deteriorated and the epidemic resumed as a second wave, which is not as strong as the first one.

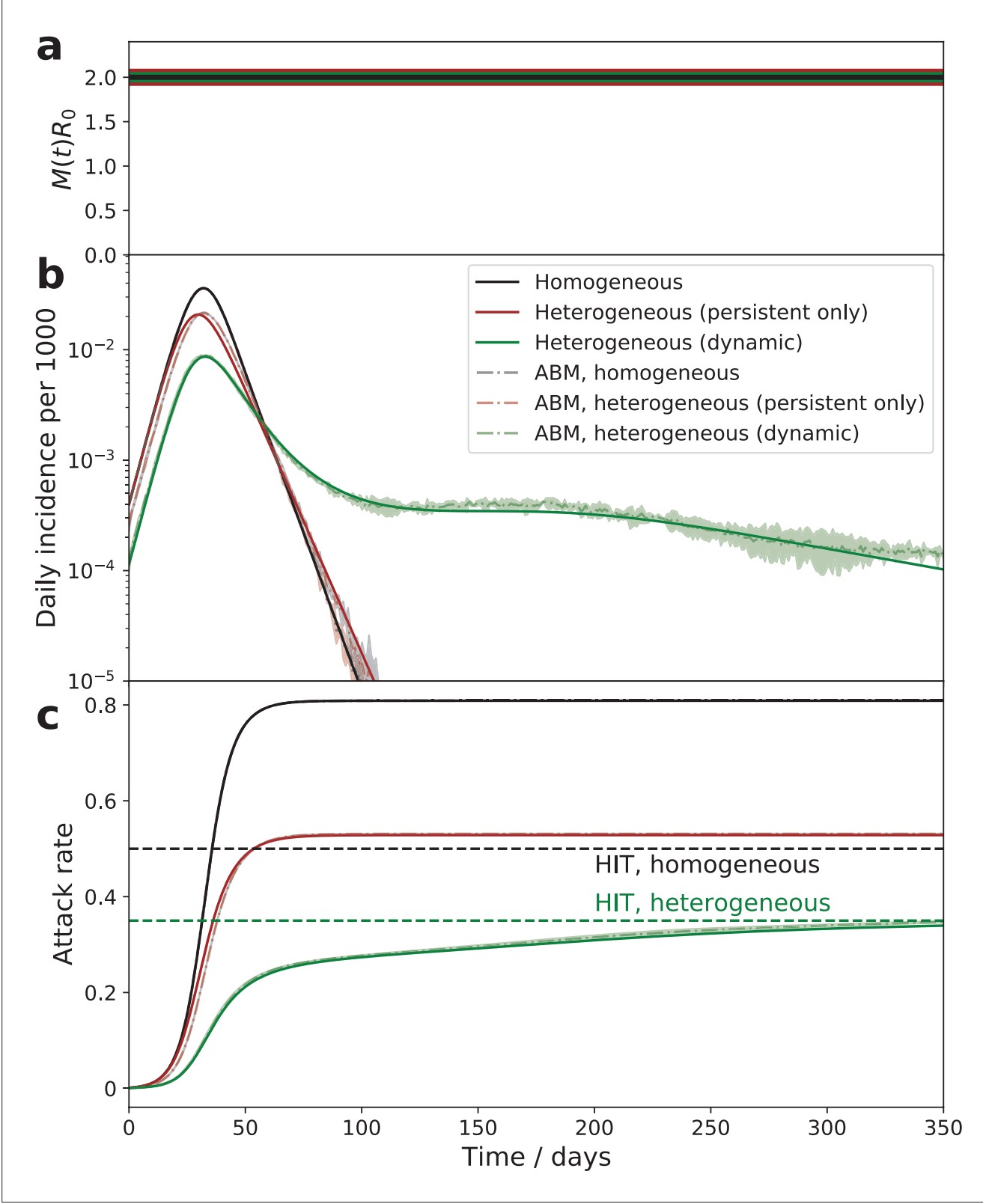

**Figure 3.** Comparison of the epidemic dynamics in three models. The mitigation profile (**a**), the daily incidence (**b**), and the cumulative attack rate (**c**). The mitigation profile (**a**), the daily incidence (**b**), and the cumulative attack rate (**c**) in the SIR model with homogeneous population (black curves), the model with persistent population heterogeneity (brown curves), and our SSA model with dynamic heterogeneity (green curves). While parameters in all three models correspond to the same herd immunity threshold (HIT), the behavior is drastically different. In the persistent model, the epidemic quickly

*Figure 3 continued on next page*

*Figure 3 continued*
overshoots above HIT level. In the case of dynamic heterogeneity, the initial wave is followed by a plateau-like behavior with slow relaxation towards the HIT. Note an excellent agreement between the quasi-homogeneous theory described by *Equation 4*; *Equation 5*; *Equation 6* (solid lines) and an agent-based model with 1 million agents whose stochastic activity is given by *Equation 1* (shaded area = the range of three independent simulations).

Since the late-stage evolution in our model is characterized by a long relaxation time $\tilde{\tau}$, the possibility of waning of individual biological immunity or escape mutations of the pathogen accumulated over certain (presumably, also long) time $\tau_b$ becomes a relevant effect. It can be incorporated as an additional relaxation term $(1 - S)/\tau_b$ in *Equation 5*. The analysis of our equations, modified in this way, shows that the universal attractor leads to a fixed point corresponding to the endemic state. That point is located somewhat below the heterogeneity-modified HIT and characterized by a finite residual incidence rate $(1 - S_\infty)/\tau_b$ and, respectively, by finite values of $I$ and $h$. Here, $S_\infty$ is the susceptible population fraction in the endemic state, which is close to but somewhat higher than that at the onset of the herd immunity. A similar endemic steady state exists in most classical epidemic models (see *Keeling and Rohani, 2011* and references therein). However, in those cases, epidemic dynamics would not normally lead to that point due to overshoot. Instead, these models typically predict a complete extinction of the disease when the prevalence drops below one infected individual. This may happen before herd immunity is lost due to waning biological immunity and/or replenishment of the susceptible population (e.g., due to births of immunologically naive individuals). That is not the case when time-dependent heterogeneity is included. Furthermore, in contrast to classical models, even in closed and reasonably small populations our mechanism would lead to an endemic state rather than pathogen extinction.

Note that for most pathogens the endemic point is not fixed, but instead is subjected to periodic seasonal forcing in $M(t)$. This leads to annual peaks and troughs in the incidence rate. Our model is able to describe this seasonal dynamics as well as the transition towards it for a new pathogen (see *Figure 5*). It captures the important qualitative features of seasonal waves of real pathogens, for example, the three endemic coronavirus families studied in *Neher et al., 2020*. They are (i) sharp peaks followed by a prolonged relaxation towards the annual minimum and (ii) a possibility of multi-annual cycles due to parametric resonance.

To understand the nature of the overall epidemic dynamics, we focus on the behavior of variables $J(t)$ and $h(t)$. Their evolution is described by *Equation 4*, *Equation 6* with $R^* = R_0 M(t) S(t)^\lambda$ playing the role of a driving force. As a result of depletion of the susceptible population, the driving force is gradually reduced, and the dynamics converges towards a slow evolution along the universal attractor shown as a black dotted trajectory in $(h, J)$ coordinates at the inset to *Figure 5*. For initial conditions away from that trajectory (say, $J \approx 0$, $h = 0$), linear stability analysis indicates that the epidemic dynamics has a damped oscillatory behavior manifesting itself as a spiral-like relaxation towards the universal attractor. A combination of this spiral dynamics with a slow drift towards the endemic state gives rise to the overall trajectory shown as the solid green line in the inset to *Figure 5*. The periodic seasonal forcing generates a limit cycle about the endemic point (small green ellipse around the red point).

More generally, any abrupt increase of the effective reproduction number, for example, due to a relaxed mitigation, seasonal changes, etc., would shift the endemic fixed point up along the universal attractor. According to *Equation 4*; *Equation 5*; *Equation 6* this will once again trigger a spiral-like relaxation. It will manifest itself as a new wave of the epidemic, such as the secondary waves in *Figure 4b*.

## Application to COVID-19 in the USA

In addition to stochastic changes in social activity, multiple other factors are known to affect the epidemic dynamics: government-imposed mitigation, knowledge-based adaptation of social behavior, seasonal forces, vaccinations, emergence of new variants, etc. Constructing and calibrating a model taking into account all of these factors is well beyond the scope of this study. A principled way of integrating the effects of mitigation and knowledge-based adaptation is to use average mobility data. By their nature, these data capture population-wide trends in social activity, while averaging out individual-level stochasticity. In *Figure 6*, we show historic Google Mobility Data for retail and recreation in four major regions of the USA: Northeast, Midwest, South, and West (*China Data Lab*

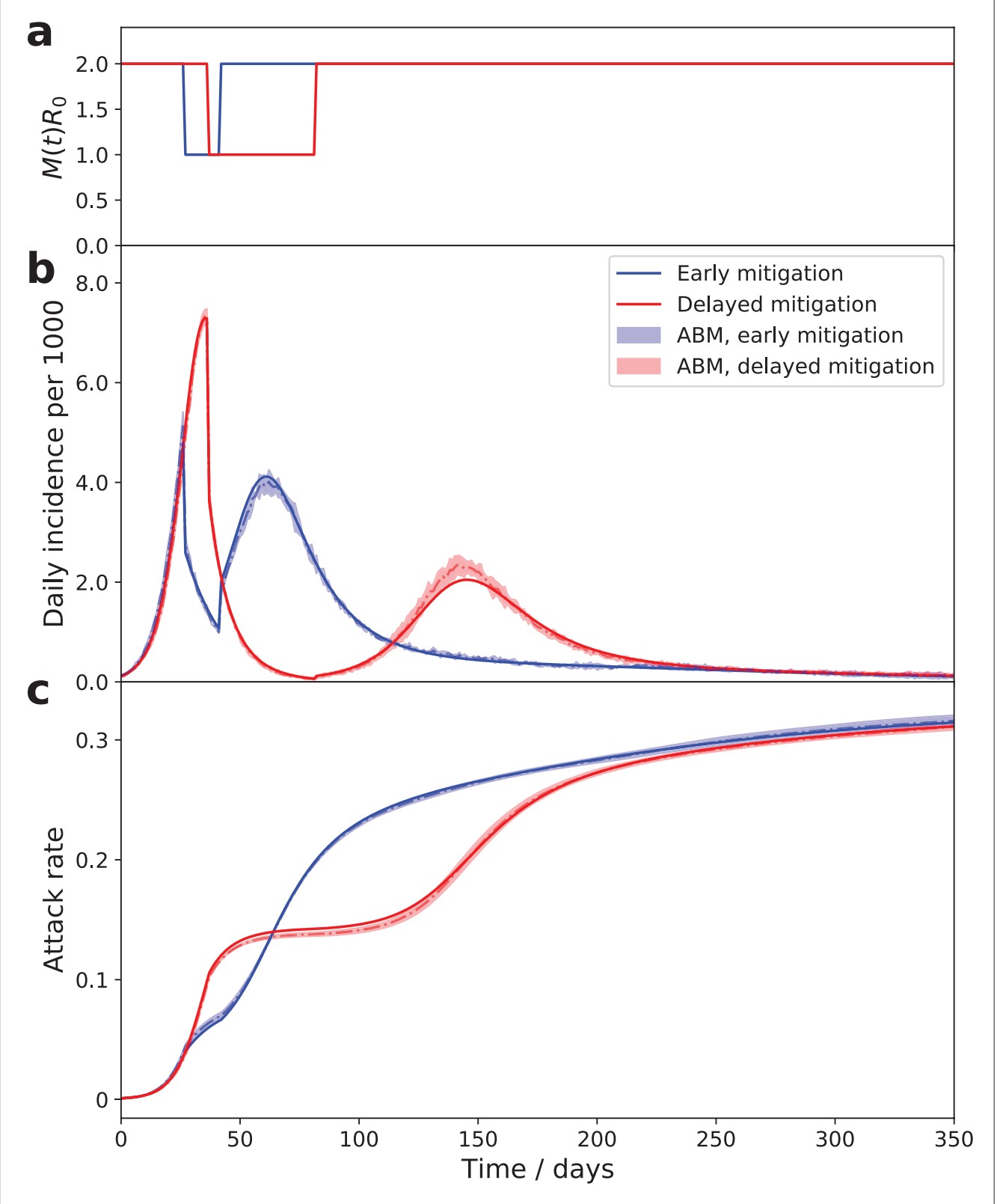

**Figure 4.** The time course of an epidemic with enhanced mitigation during the first wave. (**a**) shows the $M(t)R_0$ progression for two different strategies. In both cases, the enhanced mitigation leads to a 50% reduction of $M(t)R_0$ from 2 to 1. In the first scenario (early mitigation, blue curves), the reduction lasted for only 15 days starting from day 27. In the second scenario (delayed mitigation, red curves), the mitigation was applied on day 37 and lasted for 45 days. (**b and c**) show daily incidence and cumulative attack rates for both strategies. As predicted, differences in the initial mitigation had no

*Figure 4 continued on next page*

*Figure 4 continued*

significant effect on the epidemic in the long run: the two trajectories eventually converge towards the universal attractor. However, early mitigation allows the peak of the infection to be suppressed, potentially reducing stress on the healthcare system. A delayed mitigation gives rise to a sizable second wave.

*Dataverse, 2021*). These data exhibit pronounced effects of government-imposed mitigation and knowledge-based adaptation of the population during spring-early summer of 2020. In contrast, there is only a modest and slow variation in the mobility from mid-July 2020 through mid-February 2021 (shaded area) across all four regions. This variation is generally consistent with regular seasonal effects and lacks any signs of the drastic and fast changes similar to those observed in the early stages of the epidemic. Hence, this time interval is optimal for testing the predictions of our theory without embarking on calibration of a full-scale active mitigation model. Furthermore, this time window also excludes the effects of mass vaccinations and the introduction of COVID-19 variants of concern (*CDC,*

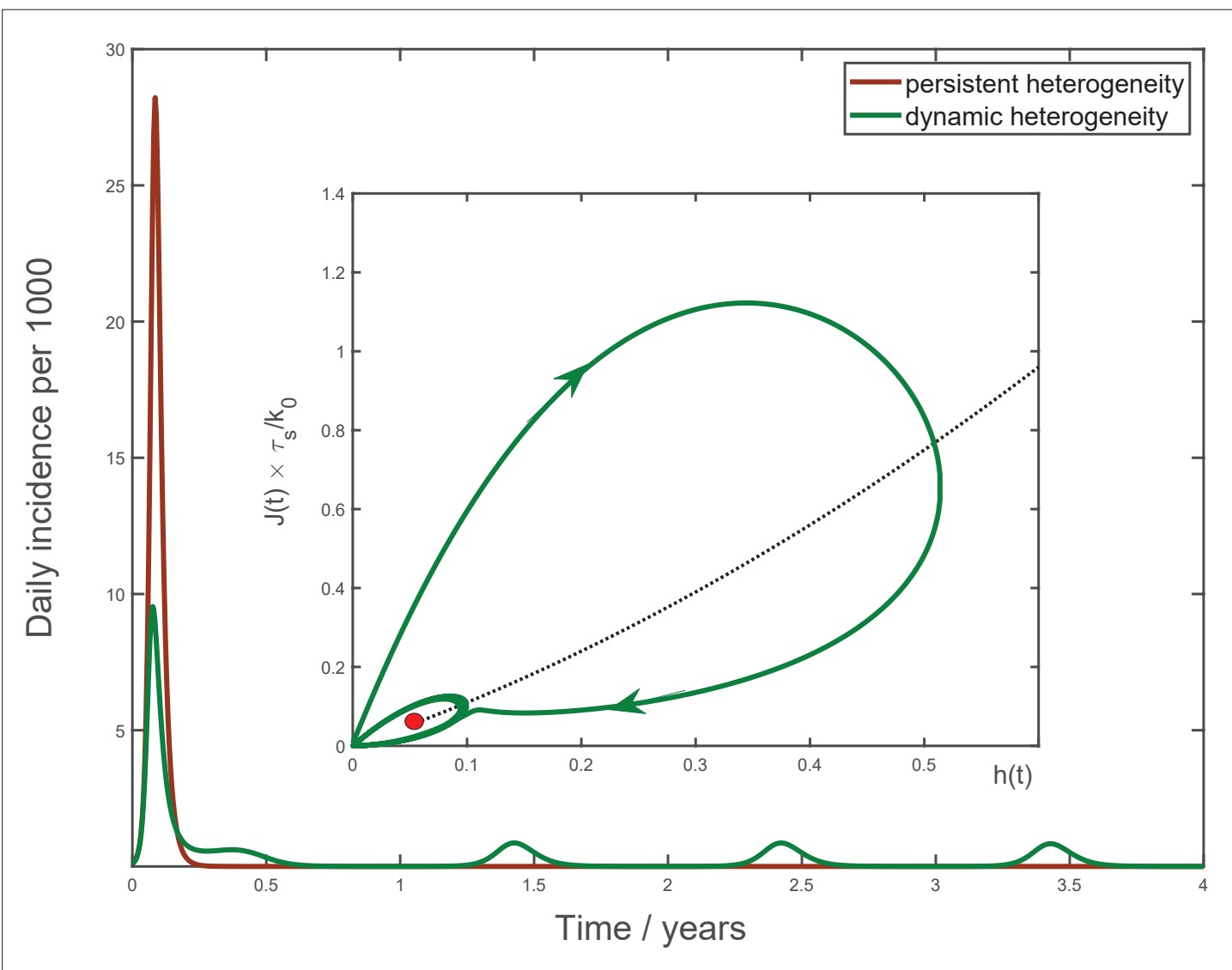

**Figure 5.** Multiyear dynamics of a hypothetical new pathogen. Effects of waning biological immunity with characteristic time $\tau_b = 5$ years, and seasonal forcing are included (see Appendix 4 for details). In the case of persistent heterogeneity without temporal variations of social activity (brown solid line), the infection becomes extinct following the initial wave of the epidemic. In contrast, dynamic heterogeneity leads to an endemic state with strong seasonal oscillations (green line). Inset: the epidemic dynamics in the $(J, h)$ phase space. The black dotted line corresponds to the universal attractor trajectory, manifested, for example, as a plateau in green line in *Figure 3b*. The attractor leads to the endemic state (red point).

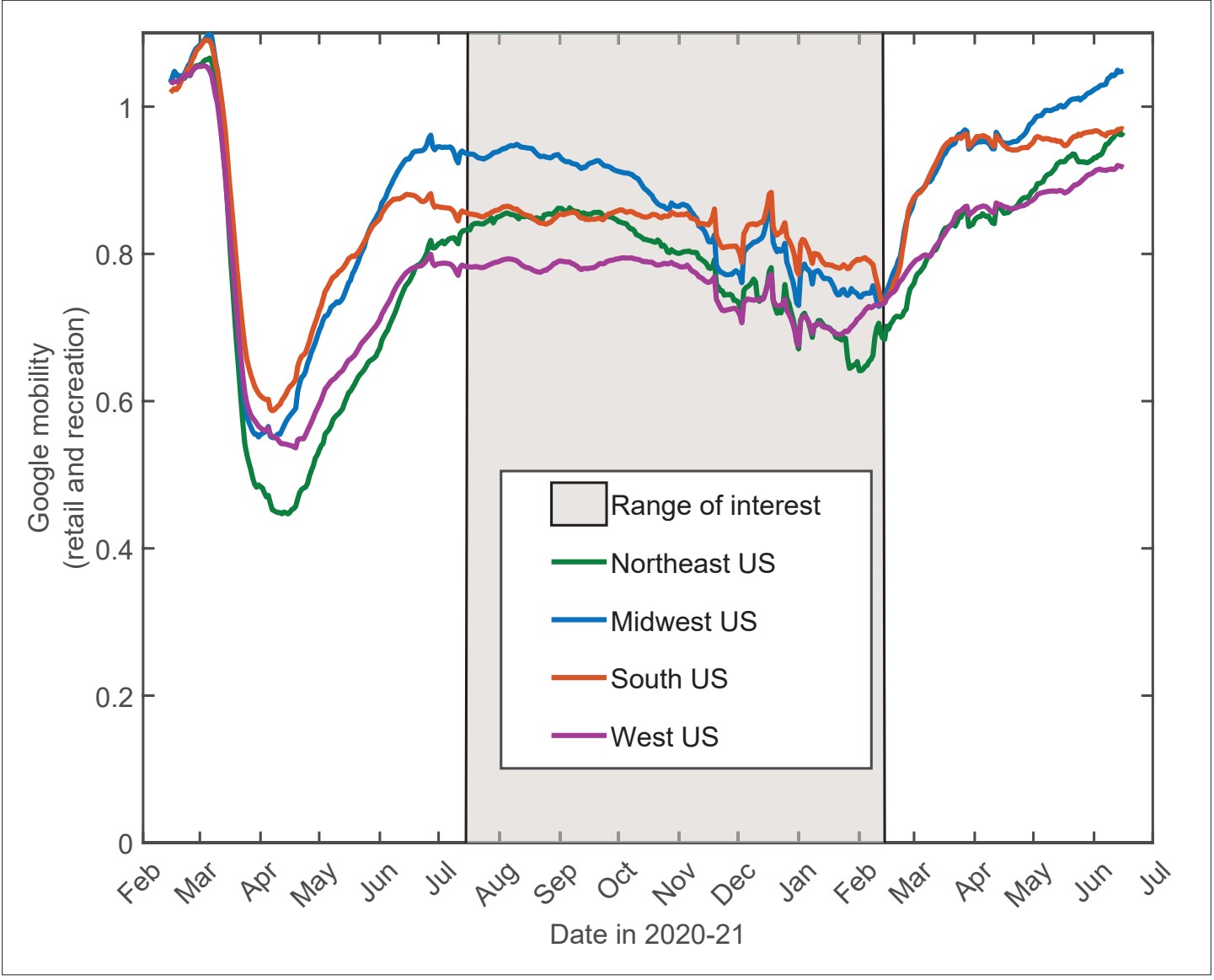

**Figure 6.** 14-day moving average of Google Mobility Data (retail and recreation) in four US regions (*China Data Lab Dataverse, 2021*). Note that the early epidemic was associated with wide and fast swings in mobility due to government-imposed mitigation and adaptive response of the population. In contrast, there is only modest and slow variation in the mobility from mid-July 2020 through mid-February 2021. This range of dates (shaded) is of interest since it allows one to directly test our theory without accounting for knowledge-based adaptation of the population.

*2021*), which became relevant after February/March 2021. Below we present a proof-of-principle demonstration that the progression of the COVID-19 epidemic from July 2020 until February 2021 in all four regions can indeed be well described by our theory.

The time dependence of daily deaths per capita (a reliable, albeit delayed measure proportional to the true attack rate) is shown in *Figure 7b and c* for each of the regions and fitted by our model with $k_0 = 0.4$, $\tau_s = 30$ days, $\kappa = 2$, together with IFR assumed to be 0.5%. This IFR value was estimated by comparing reported COVID-related deaths in the USA to two independent seroprevalence surveys (*Anand et al., 2020*; *Angulo et al., 2021*). We assume that $M(t)$ in the USA between June 2020 and February 2021 was affected primarily by seasonal dynamics. This is reflected in the simple mitigation profile $R_0 M(t)$ shown in *Figure 7* featuring a gradual seasonal increase of the reproduction number during the fall-winter period. Thus, this wave in each of the regions was triggered by the seasonal changes in transmission. According to our model, this wave was stabilized in mid-winter due to the population reaching the TCI state. There is a good agreement between our model and the empirical

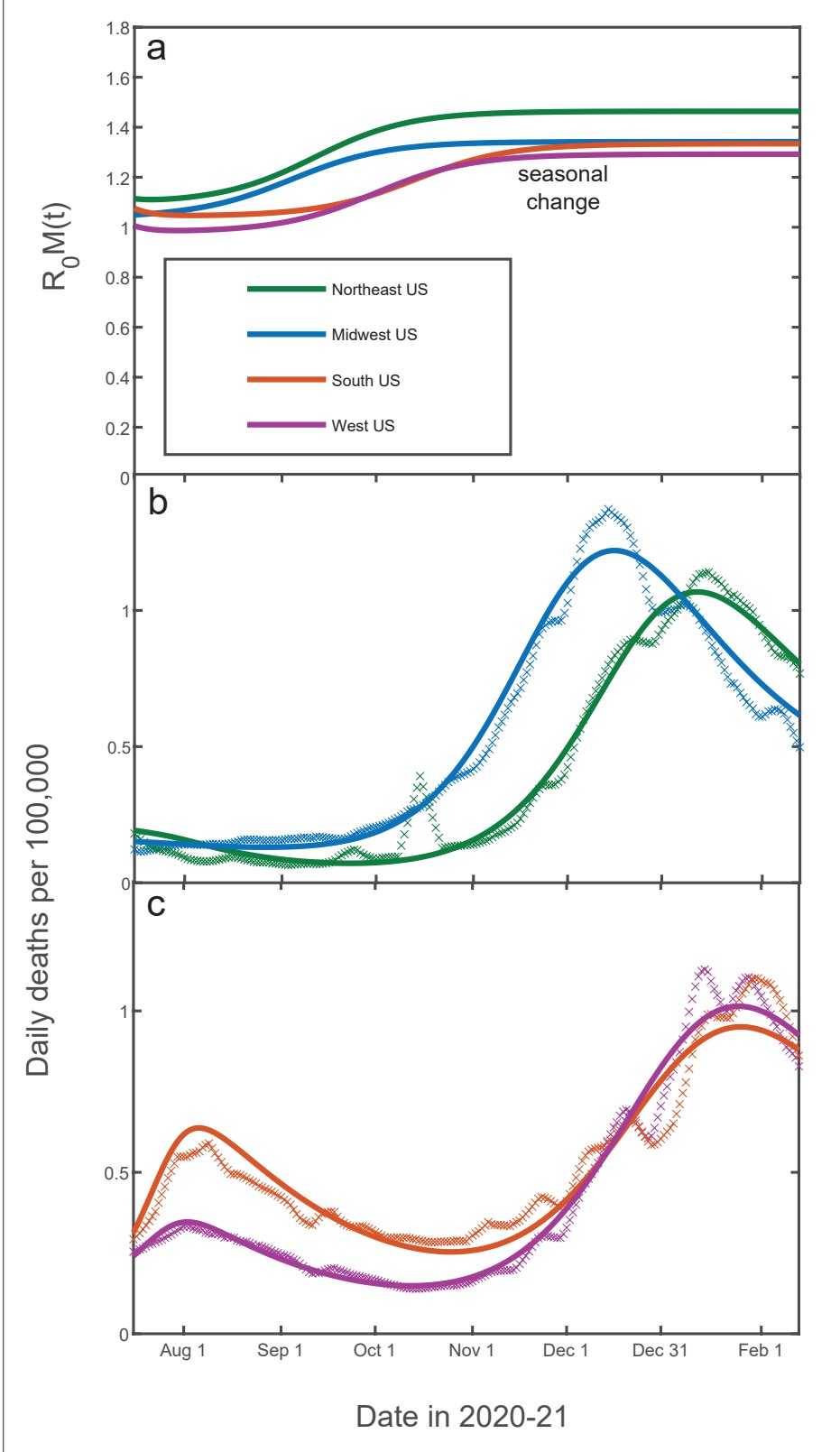

**Figure 7.** Fitting of the empirical data on COVID-19 epidemic in Northeast (green), Midwest (blue), West (purple), and South (orange) of the USA. The time range corresponds to the shaded region in *Figure 6*. The best-fit profiles of $R_0 M(t)$ within this range (panel **a**) are shaped only by seasonal changes. The time dependence of daily deaths per capita for the Northeast and Midwestern regions of the USA (panel **b**) as well as for Southern and Western

*Figure 7 continued on next page*

*Figure 7 continued*

regions (panel **c**). Data points represent reported daily deaths per 100,000 of population for each of the regions. Solid lines are the best theoretical fits with our model (see Appendix 5 for details of the fitting procedure).

data for all four regions. Note that the shape of the seasonal epidemic wave is determined by the relative change of $R_0 M(t)$ between summer and winter, or, equivalently, by the height of the peak itself. Analysis of *Equation 4*; *Equation 5*; *Equation 6* shows that, for a given height, the peak is shaped by three underlying model parameters: $\gamma$, $k_0$, and $\tau_s$. Since one of them, $\tau_s$, could not be determined from independent studies, we checked the sensitivity of our model to the choice of that timescale. It was found that the best-fit values of $\tau_s$ range from 20 to 55 days for different US regions, and that the overall agreement remains very good for any value within that range (see Appendix 5 for further details).

Finally, we performed a critical test of the predictive power of our theory. To do that, the empirical data in Midwest region have been fitted up to November 15, 2020, and the epidemic dynamics beyond that date has been projected by our SSA model. As shown in *Figure 8*, this procedure gives

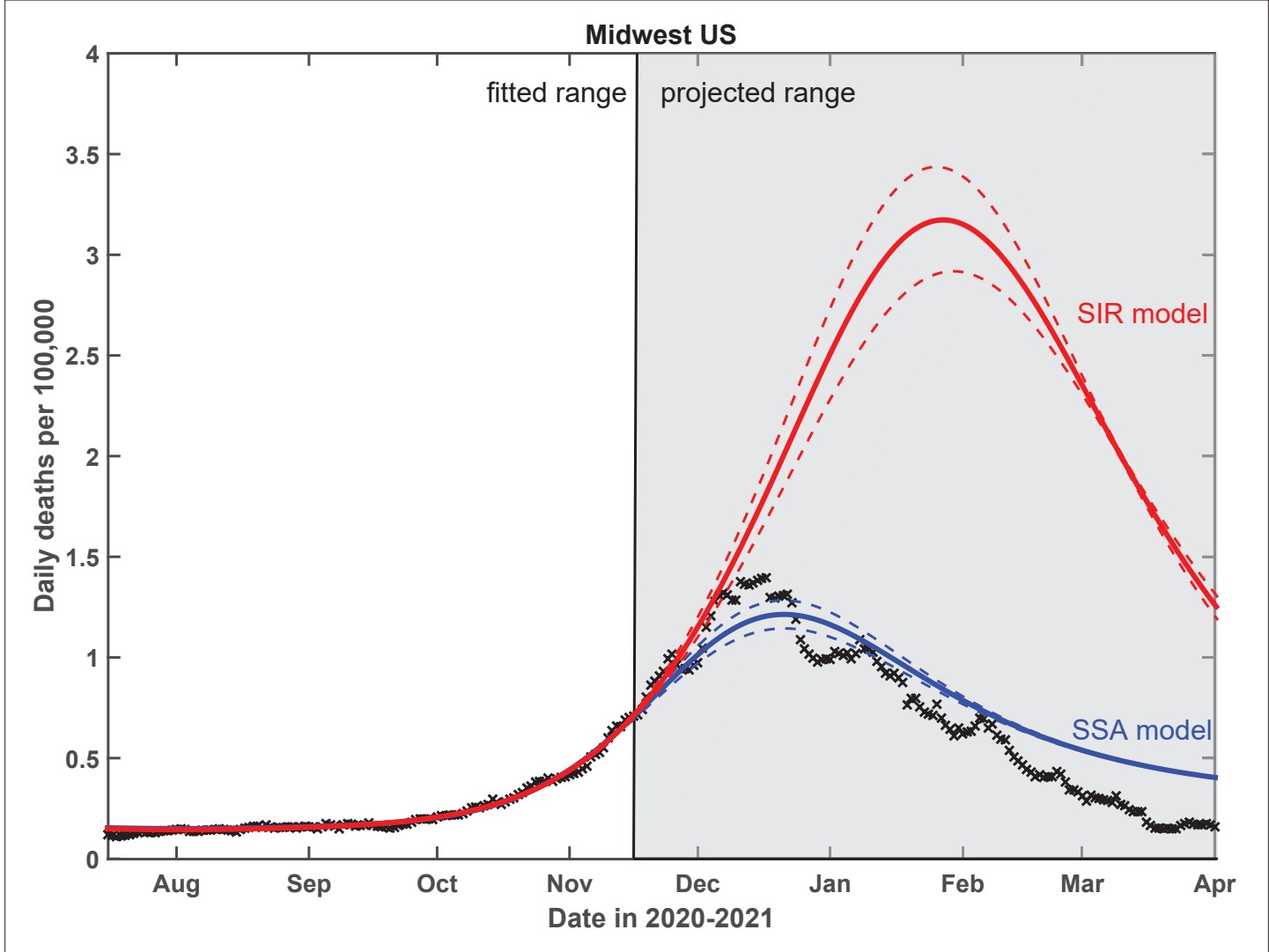

**Figure 8.** Test of the predictive power of the stochastic social activity (SSA) model developed in this work. Daily deaths data in the Midwest region of the USA have been fitted up to November 17, 2020. The epidemic dynamic beyond that date has been projected by our model (blue). One observes a good agreement between this prediction and the reported data (crosses). In contrast, the classical susceptible-infected-removed (SIR) model (red) substantially overestimates the height of the peak and projects it at a much later date than had been observed. Solid lines represent the best-fit behavior for each of the models, while dotted lines indicate the corresponding 95% confidence intervals.

a very good prediction of the overall seasonal wave, based only on its onset behavior. In contrast, use of the traditional SIR model leads to an almost threefold overestimate of the height of the peak, with predicted timing about a month later than is observed. To fit the data with the standard SIR model, we forced $h(t) = 0$ at all times and set $\lambda = 1$. The fitting procedure and the range of fitted dates for the SIR model was identical to that of the SSA model. We chose to show the Midwest region in the main text partly because a part of this region (the state of Illinois) was the subject of our previous publication (*Wong et al., 2020*). The fits to all four US regions are shown in *Appendix 5—figure 2*. The timing of the peak of the wave in all four regions is in closer agreement with the SSA model than the SIR model. The same is true for the height of the peak except for the South region, where it is somewhere in between the predictions of these two models.

## Discussion

In conclusion, we have proposed a new theory integrating the stochastic dynamics of individual social activity into traditional epidemiological models. Our SSA model describes the so-called 'zero intelligence' limit in which there is no feedback from the epidemic dynamics to social activity, for example, mediated by the news. Hence, our approach is complementary to knowledge-based models of *Epstein et al., 2008*; *Funk et al., 2009*; *Fenichel et al., 2011*; *Bauch, 2013*; *Rizzo et al., 2014*; *Weitz et al., 2020*; *Arthur et al., 2021*. The SSA in our approach is described by the CIR model (*Cox et al., 1985*), which captures the following important properties: (i) the activity cannot be negative; (ii) for any given individual, it reverses towards its long-term average value; and (iii) it exhibits gamma-distributed short-term overdispersion (aka superspreading) (*Lloyd-Smith et al., 2005*; *Endo et al., 2020*; *Sun et al., 2021*). We mapped the overall epidemic dynamics featuring heterogeneous time-varying social activity onto a system of three differential equations, two of which generalize the traditional SIR model. The third equation describes the dynamics of the heterogeneity variable $h(t)$, driven up by the current strength of infection $J(t)$ and relaxing back to zero due to variable social activity.

The emergent property of our theory is the new long timescale of the order of $\tau_s/k_0$ governing the relaxation towards either the herd immunity or the endemic state of the pathogen. For parameters relevant for COVID-19 epidemic, this timescale is approximately five times longer than the relaxation time constant for social activity $\tau_s$. This emergent timescale might be of relevance to public health measures as it describes when the epidemic is reaching a sustainable plateau and for how long this plateau is expected to last.

The long-term dynamics of our model is in striking contrast to traditional epidemiological models, generally characterized by a large overshoot above the HIT leading to a likely extinction of new pathogens. Our theory provides a plausible explanation for the long plateaus observed in real-life epidemics such as COVID-19. It also provides a qualitative description of transient suppression of individual epidemic waves well below the HIT (*Tkachenko et al., 2021*). In particular, this mechanism explains how the winter 2020/21 waves of the COVID-19 epidemic in the USA were suppressed in the absence of a noticeable reduction in the population mobility.

## Data availability

All code needed to reproduce results of our Agent Based Model and fits of the epidemic dynamics in US regions is available on Github at https://github.com/maslov-group/COVID-19-waves-and-plateaus (*Maslov and Wang, 2021*; copy archived at swh:1:rev:1e03ff622f16b85515e7162eab77ebd8e4efd30a).

## Acknowledgements

This work was supported by the University of Illinois System Office, the Office of the Vice-Chancellor for Research and Innovation, the Grainger College of Engineering, and the Department of Physics at the University of Illinois at Urbana-Champaign. This research was partially done at and used resources of the Center for Functional Nanomaterials, which is a U.S. DOE Office of Science Facility, at Brookhaven

National Laboratory under Contract No. DE-SC0012704. The work of SM was supported in part by the NSF award 2107344.

## Additional information

### Funding

| Funder | Grant reference number | Author |
|---|---|---|
| U.S. Department of Energy | DE-SC0012704 | Alexei V Tkachenko |
| University of Illinois at Urbana-Champaign | University of Illinois System Office,Office of Vice-Chancellor,the Grainger College of Engineering | Sergei Maslov Tong Wang Ahmed Elbana George N Wong Nigel Goldenfeld |
| National Science Foundation | 2107344 | Sergei Maslov |

The funders had no role in study design, data collection and interpretation, or the decision to submit the work for publication.

### Author contributions
Alexei V Tkachenko, Conceptualization, Data curation, Formal analysis, Investigation, Methodology, Software, Validation, Visualization, Writing – original draft; Sergei Maslov, Project administration, Formal analysis, Investigation, Project administration, Software, Validation, Visualization, Writing – original draft, ; Tong Wang, Conceptualization, Formal analysis, Investigation, Software, Validation, Visualization; Ahmed Elbana, Conceptualization, Formal analysis, Validation, ; George N Wong, Formal analysis, Software, Validation, ; Nigel Goldenfeld, Project administration, Formal analysis, Investigation, Project administration, Validation,

### Author ORCIDs
Alexei V Tkachenko https://orcid.org/0000-0003-1291-243X
Sergei Maslov https://orcid.org/0000-0002-3701-492X

### Decision letter and Author response
Decision letter https://doi.org/10.7554/eLife.68341.sa1
Author response https://doi.org/10.7554/eLife.68341.sa2

## Additional files

### Supplementary files
• Transparent reporting form

### Data availability
All code needed to reproduce results of our Agent Based Model and fits of the epidemic dynamics in US regions is available on Github https://github.com/maslov-group/COVID-19-waves-and-plateaus copy archived at https://archive.softwareheritage.org/swh:1:rev:1e03ff622f16b85515e7162eab77ebd8e4efd30a.

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

## Appendix 1

### Epidemic dynamics with dynamic heterogeneity

Let $a_i(t)$ be the measure of an individual's social activity proportional to the frequency and the intensity of this person's close contacts with other people around time $t$. We refer to it as (social) susceptibility to infection, but it also determines one's potential to infect others. In particular, the infectivity of a person infected at time $t_i^*$ at a later time $t_i^* + \tau$ is given by

$$\beta_i(t_i^* + \tau) = C_i(\tau)a_i(t^* + \tau) \tag{S1}$$

Here, $C_i(\tau)$ is this person's contagiousness at time $\tau$ after the infection.

Let $j(t)$ be the fraction of infected individuals, weighted proportionally to their current infectivity level, and $M(t)$ be the mitigation factor that reflects governmental and social response to the epidemic, seasonal effects, etc. Their product, $J(t) = M(t)j(t)$, is the force of infection, that is, a hypothetical incidence rate in a fully susceptible homogeneous population with $\bar{\alpha} = 1$. Within the heterogeneous (but well-mixed) age-of-infection model, the current value of $j(t)$ is given by

$$j(t) = \left\langle \beta_i(t - t_i^*) \right\rangle_i = \left\langle \int_0^\infty C_i(\tau)a_i(t)a_i(t - \tau)S_i(t - \tau)J(t - \tau)d\tau \right\rangle_i \tag{S2}$$

Here, $S_i(t - \tau)$ is the state of an individual (1 if susceptible, 0 otherwise), and $a_i(t - \tau)S_i(t - \tau)J(t - \tau)$ is the probability of this individual to get infected at time $t - \tau$. To calculate the infectivity-weighted fraction of all individuals $j(t)$, that probability needs to be multiplied by this person's contagiousness $C_i(\tau)$ and social activity level $a_i(t)$ at time $t$. It is then averaged over all times since infection $\tau$ and the entire population. Since the strength of infection $J(t)$ is by definition proportional to $j(t)$, we obtain the quasi-homogeneous renewal equation:

$$j(t) = \int_0^\infty K(t, \tau)R_e(t - \tau)j(t - \tau)d\tau \tag{S3}$$

Here, the effective reproduction number $R_e$ and the probability density of the generation interval $\tau$, $K(\tau)$, are given by

$$R_e(t) = M(t) \left\langle S_i(t) \int_0^\infty a_i(t)a_i(t + \tau)C_i(\tau)d\tau \right\rangle_i \tag{S4}$$

$$K(t, \tau) = \frac{\left\langle S_i(t)a_i(t)a_i(t + \tau)C_i(\tau) \right\rangle_i}{\left\langle S_i(t) \int_0^\infty a_i(t)a_i(t + \tau)C_i(\tau)d\tau \right\rangle_i} \tag{S5}$$

### Stochastic social activity model

It is well known that social interactions are 'bursty.' That is to say, individual social activity has both (nearly) permanent and significant time-dependent contributions:

$$a_i(t) = \bar{\alpha}_i + \delta a_i(t) \tag{S6}$$

Without loss of generality, we set the population-averaged permanent and instantaneous susceptibility to 1: $\langle a_i(t) \rangle_i = \langle \bar{\alpha}_i \rangle_i = 1$. Beyond its average value, the overall statistics of instantaneous $\bar{\alpha}(t)$ is properly defined only if that quantity is average over specified time window $\delta t$. Naturally, its variation will gradually decrease as the time widow increases.

The individual reproductive number, $R_i$, for COVID-19 epidemics is (in)famously overdispersed. This is a result of superspreading when a majority of secondary infections are caused by a small fraction of index cases. The overdispersion reflects (i) variation of peak contagiousness level among individuals and (ii) dispersion of $a_i(t)$, which is effectively averaged over a timescale of the peak infection period (approximately 2 days).

Importantly, according to *Equation S4* the reproductive number depends on correlations of $a_i$ across a timescale of a single generation interval (on average, 4–5 days for COVID 19). Thus, any variations in $a_i(t)$ that do not persist over that timescale would be averaged out. Here, we introduce a simple model to account for temporal variation of social activity. This model is well known in mathematical finance as the CIR model (*Cox et al., 1985*) and has been studied in probability theory

since the 1950s (**Feller, 1951**). It captures the following important properties of the social activity: (i) the social activity cannot be negative; (ii) for any given individual, it reverses towards its long-term average value; and (iii) it exhibits gamma-distributed short-term overdispersion (aka superspreading) (**Lloyd-Smith et al., 2005**; **Endo et al., 2020**; **Sun et al., 2021**). In the model, $a_i$ of a given individual may vary on a short timescale and relax to its persistent value over a certain relaxation time, $\tau_s$.

$$\dot{a}_i = \frac{\bar{\alpha}_i - a_i}{\tau_s} + \eta_i(t) \tag{S7}$$

In other words, we assume that $a_i(t)$ follows a stochastic differential equation (SDE) with a zero mean Gaussian noise $\eta(t)$ term and correlation function $\langle \eta_i(t)\eta_i(t')\rangle = \frac{2a_i(t)}{\tau_s k_0}\delta(t - t')$. Here, $\delta(t - t')$ is the Dirac delta function. This SDE describes the diffusion process in the $a_i$-space with the diffusion coefficient proportional to $a_i$. The evolution of the subpopulation with a given value of persistent activity $\bar{\alpha}$ in that space is given by the following Fokker–Plank equation:

$$\dot{\Psi}_{\bar{\alpha}}(a, t) = \frac{1}{k_0 \tau_s}\frac{\partial^2\left(a\Psi(a,t)_{\bar{\alpha}}\right)}{\partial a^2} + \frac{1}{\tau_s}\frac{\partial\left((a-\bar{\alpha})\Psi_{\bar{\alpha}}(a,t)\right)}{\partial a} \tag{S8}$$

The steady-state solution to this equation gives a probability density function (pdf) for $a$, which turns out (see **Feller, 1951**) to be the commonly used gamma distribution:

$$\Psi_{\bar{\alpha}}(a, t) = f_{\bar{\alpha}}(a) = \frac{a^{\bar{\alpha}k_0 - 1}e^{-k_0 a}}{\bar{\alpha}^{\bar{\alpha}k_0}\Gamma(\bar{\alpha}k_0)} \tag{S9}$$

Note that the statistics of superspreading events is commonly modeled assuming the very same distribution for individual reproduction number, $R_i$. This gives a strong empirical support to the chosen model, in particular to the choice to let the diffusion coefficient be proportional to $\bar{\alpha}$. It also allows us to partially calibrate the model. The reported dispersion parameter associated with superspreading events for COVID-19 is in the range of 0.1–0.3 (**Endo et al., 2020**; **Sun et al., 2021**). Note, however, that our parameter $k_0$ is expected to be larger than $k$, that is, has a smaller dispersion. This is because variations of $a(t)$ over the timescale shorter than a single generation interval would be averaged out according to **Equation S10**, while the superspreading statistics effectively probes it over a shorter time interval of the infectivity peak in a single individual. The latter could be further enhanced by a variation of average transmission probability for an infectious individual, for example, due to biological factors.

In our model, we account for individual variations of the average social activity $\bar{\alpha}$ and assume that it also obeys a gamma distribution, $p(\bar{\alpha}) \sim \bar{\alpha}^{\kappa-1}e^{-\kappa\bar{\alpha}}$. Throughout this study, we set $\kappa = 2$ and $k_0 = 0.4$ as justified in the main text. The pdfs of the corresponding distributions of the instantaneous, $a_i$, and persistent, $\bar{\alpha}_i$, social activities are shown in **Appendix 1—figure 1**, as black and blue curves, respectively.

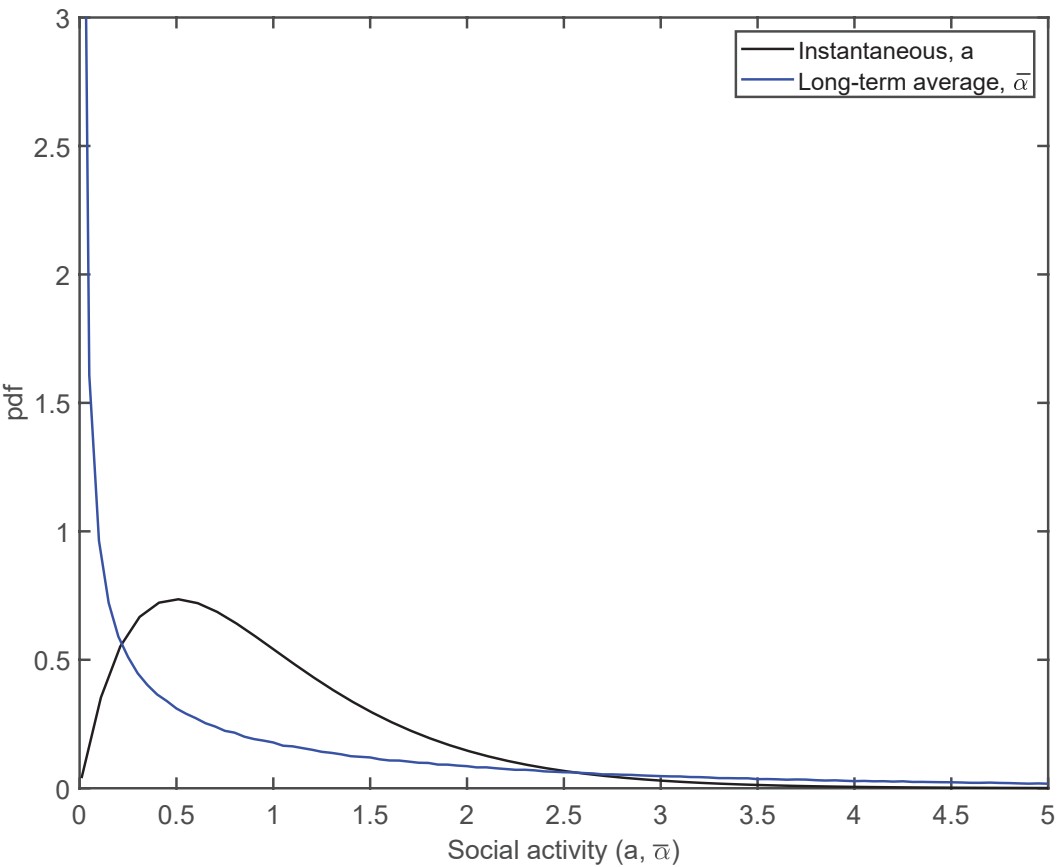

**Appendix 1—figure 1.** The pdfs of distributions of the instantaneous (black) and persistent (blue) social activities.

## Effect of dynamic heterogeneity on basic reproduction number

It is well known that the mean reproduction number $R_0$ in a heterogeneous population depends on the second moment of the distribution of $\bar{\alpha}$ (in network epidemic models, it is related to the individual degree). However, there is an important modification to that result for time-dependent $a(t)$:

$$R_0 = \int_0^\infty \langle a_i(t)\beta_i(t+\tau)\rangle_i \, d\tau = R\langle\bar{\alpha}_i^2\rangle_i + \int_0^\infty \langle C_i(\tau)\delta a_i(t)\delta a_i(t+\tau)\rangle_i \, d\tau \tag{S10}$$

Here, $R = \langle \int C_i(\tau)d\tau \rangle_i$ is the net infection transmission probability of an average person. Above we assumed statistical independence of $C_i$ and $\bar{\alpha}_i$ as well as time independence of $R_0$, which guarantees vanishing of all terms linear in $\delta a_i$. From the previous equation, one gets

$$R_0 = R\langle\bar{\alpha}_i^2\rangle_i + \langle\delta a_i^2(t)\rangle_i \int_0^\infty \langle C_i(\tau)\rangle_i e^{-\tau/\tau_s}d\tau = R\left(\langle\bar{\alpha}_i^2\rangle_i + \mu k_0^{-1}\right) \tag{S11}$$

Here, we neglected any correlation between the individual contagiousness $C_i(t)$ and variations in social activity $\delta a_i(t)$. We also used the fact that in the CIR model (**Feller, 1951**; **Cox et al., 1985**) autocorrelations decay exponentially, $\langle\delta a_i(t)\delta a_i(t+\tau)\rangle_i = \langle\delta a_i^2(t)\rangle_i e^{-\tau/\tau_s}$, and that $\langle\delta a_i^2(t)\rangle_i = k_0^{-1}$. The factor $\mu$ is related to the Laplace transform of the average contagiousness profile, $K_0(\tau) = \langle C_i(\tau)\rangle_i/R$

$$\mu = \int_0^\infty K_0(\tau)e^{-\tau/\tau_s}d\tau \tag{S12}$$

Note that, according to **Equation S5**, the generation interval pdf $K(\tau)$ is close, but not identical, to $K_0(\tau)$:

$$K(\tau) = \left(1 + \frac{e^{-\tau/\tau_s} - \mu}{k_0 \langle \bar{\alpha}_i^2 \rangle_i + \mu}\right) K_0(\tau) \tag{S13}$$

For instance, consider a case of the SIR infection dynamics in which every individual transitions from the infectious to the removed states at rate $\gamma_0$. In most versions of the SIR model (either heterogeneous or homogeneous), the mean generation interval is $1/\gamma_0$. This is not quite the case for the SSA model with stochastic social dynamics. There is a correction to the mean generation interval, $1/\gamma$, due to time variations in $a_i(t)$:

$$\frac{1}{\gamma} = \frac{1}{\gamma_0} \frac{\langle \bar{\alpha}_i^2 \rangle_i + \mu^2 k_0^{-1}}{\langle \bar{\alpha}_i^2 \rangle_i + \mu k_0^{-1}} \approx \frac{1}{\gamma_0} \left(1 - \frac{1}{\gamma_0 \tau_s (1 + k_0 \langle \bar{\alpha}_i^2 \rangle_i)}\right) \tag{S14}$$

In the right-hand side of this equation, we used

$$\mu = \left(1 + \frac{1}{\gamma_0 \tau_s}\right)^{-1} \tag{S15}$$

and kept the leading corrections in $\frac{1}{\gamma_0 \tau_s}$. In the case of SIR dynamics, one can assign each person a state variable $I_i$ set to 1 when the individual is infectious and 0 otherwise. This allows us to describe the epidemic dynamics in terms of activity-weighted fraction of the infected population, $I(t) = \langle I_i a_i(t) \rangle_i / \langle a_i^2 \rangle_i$. Note that variable $j(t)$ and hence the strength of infection are proportional to it:

$$J(t) = M(t)j(t) = \gamma R_0 M(t) I(t) \tag{S16}$$

## Appendix 2

### Mapping on quasi-homogeneous dynamic system

Let $S_{\bar{\alpha}}(a,t)$ be the fraction of susceptibles among the subpopulation with persistent activity level $\bar{\alpha}$ and given instantaneous activity level $a$, at time $t$. The change of the function $S_{\bar{\alpha}}(a,t)$ is driven by two effects: (i) depletion of the susceptible population due to infection and (ii) diffusion of individual in $a$-space. By substituting $\Phi_{\bar{\alpha}}(a,t) = f_{\bar{\alpha}}(a)S_{\bar{\alpha}}(t)$ into Fokker–Plank *Equation S8*, and adding the infection term with rate $-a(t)$, we obtain an evolution equation for $S_{\bar{\alpha}}$:

$$\dot{S}_{\bar{\alpha}}(a,t) = -aS_{\bar{\alpha}}(a,t)J(t) + \frac{a}{k_0\tau_s}\frac{\partial^2 S_{\bar{\alpha}}(a,t)}{\partial a^2} + \left(\frac{\bar{\alpha}-a}{\tau_s}\right)\frac{\partial S_{\bar{\alpha}}(a,t)}{\partial a} \tag{S17}$$

This equation can be solved by using the following ansatz:

$$S_{\bar{\alpha}}(a,t) = \exp\left[-Z(t)\bar{\alpha} - k_0 h(t)a\right] \tag{S18}$$

Here, $Z(t)$ is a measure of persistent heterogeneity: the larger it is, the more is the difference in depletion of susceptibles among subpopulations with different $\bar{\alpha}$, that is, various average levels of social activity. On the other hand, $h(t)$ parameterizes the transient heterogeneity within each of these subpopulations. In the long run, this type of heterogeneity disappears due to diffusion in $a$-space, thus $h(t)$ asymptotically approaches 0 as $t \to \infty$. Substituting *Equation S18* into *Equation S17* results in simple equations for both $Z(t)$ and $h(t)$:

$$\dot{h} = \frac{J(t)}{k_0} - \frac{h(t)(1+h(t))}{\tau_s} \tag{S19}$$

$$\dot{Z} = \frac{k_0 h(t)}{\tau_s} \tag{S20}$$

The renewal equation *Equation S3* for $j(t)$ completes our quasi-homogeneous description of the epidemic dynamics. However, to fully close this system of equations, one needs to express the effective reproduction number, $R_e$, in terms of the functions $M(t)$, $Z(t)$, and $h(t)$. This is done by substituting the ansatz, *Equation S18*, into *Equation S4*. We perform this calculation in two steps, by first finding the effective number $R_{\bar{\alpha}}$ for a subpopulation with average level of activity $\bar{\alpha}$, followed by averaging over persistent heterogeneity. This gives

$$R_{\bar{\alpha}} = \int_0^\infty a(\bar{\alpha} + \mu(a-\bar{\alpha}))f_{\bar{\alpha}}(a)e^{-Z(t)\bar{\alpha}-k_0 h(t)a}da = \frac{\bar{\alpha}R\left(\bar{\alpha} + \mu k_0^{-1} + h(1-\mu)\right)e^{-\tilde{Z}\bar{\alpha}}}{\left(1+h(t)\right)^2} \tag{S21}$$

Here,

$$\tilde{Z} = Z + k_0\ln(1+h) \tag{S22}$$

Note that

$$\dot{\tilde{Z}} = \frac{J(t)}{1+h(t)} \tag{S23}$$

The averaging over persistent heterogeneity, under the assumption that $\bar{\alpha}$ obeys the gamma distribution, $p(\bar{\alpha}) \sim \bar{\alpha}^{\kappa-1}e^{-\kappa\bar{\alpha}}$, yields

$$R_e(t) = M(t)\int_0^\infty R_{\bar{\alpha}}p(\bar{\alpha})d\bar{\alpha} = \frac{\chi + \left(1-\chi\right)(1+k_0 h(\mu^{-1}-1))\left(1+\kappa^{-1}\tilde{Z}(t)\right)R_0 M(t)}{\left(1+\kappa^{-1}\tilde{Z}(t)\right)^{2+\kappa}\left(1+h(t)\right)^2} \tag{S24}$$

Here,

$$\chi = \frac{1+\kappa^{-1}}{1+\kappa^{-1}+\mu k_0^{-1}} \tag{S25}$$

Similarly, we calculate $S$, which ends up having the same form as in the model with persistent heterogeneity (**Tkachenko et al., 2021**):

$$S(t) = \int_0^\infty \int_0^\infty p(\bar{\alpha}) f_{\bar{\alpha}}(a) e^{-Z(t)\bar{\alpha} - k_0 h(t)a} da d\bar{\alpha} = \frac{1}{\left(1 + \kappa^{-1} \tilde{Z}(t)\right)^\kappa} \tag{S26}$$

By comparing **Equation S24** and **Equation S26** we obtain $R_e$ in terms of $S$ and $h$:

$$R_e(t) = \frac{R_0 M(t) S^\lambda q_\chi(S,h)}{\left(1 + h(t)\right)^2} \tag{S27}$$

Here,

$$q_\chi(S,h) = (1 - \chi)\left(1 + k_0 h(\mu^{-1} - 1)\right) S^{-\chi/\kappa} + \chi S^{(1-\chi)/\kappa} \approx 1 \tag{S28}$$

$$\lambda = 1 + \frac{1 + \chi}{\kappa} = \frac{\left(1 + \kappa^{-1}\right)\left(1 + \mu k_0^{-1} + 2\kappa^{-1}\right)}{1 + \mu k_0^{-1} + \kappa^{-1}} \tag{S29}$$

Note that for most practical purposes one can set $q_\chi(S,h) = 1$. According to **Equation S27**, the effective reproduction number is explicitly suppressed by the current level of transient heterogeneity $h(t)$. This is exactly the mechanism of TCI introduced in our earlier study (**Tkachenko et al., 2021**). The parameter $\lambda$ is the 'immunity factor' described in the same study. In the case of persistent heterogeneity, $\lambda = 1 + 2/\kappa$ appears as the scaling exponent in the relationship between the effective reproduction number $R_e(t)$ and the fraction of the susceptible population $S(t)$. Our **Equation S27** generalizes that result.

**Equation S3**, **Equation S19**, **Equation S27**, **Equation S23** give a full description of the epidemic dynamics in heterogeneous system. For the particular case of the SIR model ($K(\tau) \sim e^{-\gamma\tau}$), we obtain a 3D dynamical system in terms of variables $I(t)$, $S(t)$ and $h(t)$:

$$\frac{dI}{dt} = \frac{JS^\lambda}{\left(1 + h\right)^2} - \gamma I \tag{S30}$$

$$\frac{dS}{dt} = -\frac{JS^{1+1/\kappa}}{(1 + h)} \tag{S31}$$

$$\frac{dh}{dt} = \frac{J}{k_0} - \frac{h(1 + h)}{\tau_s} \tag{S32}$$

Here, $J(t) = \gamma R_0 M(t) I(t)$, as given by **Equation S16**. **Equation S31** was derived by combining **Equation S23** and **Equation S26**. Alternatively, after substituting the result of integration of **Equation S23** into **Equation S26**, one gets the explicit formula for $S(t)$:

$$S(t) = \left(1 + \kappa^{-1} \int_{-\infty}^t \frac{J(t')dt'}{1 + h(t')}\right)^{-\kappa} \tag{S33}$$

## Appendix 3

## Waves and plateaus

According to *Equation S30*, the combined driving force of the epidemic is $R^* = R_0 M(t) S^\lambda(t)$. It includes both the effects of mitigation $M(t)$ and suppression associated with the buildup of the long-term herd immunity. First, we assume $R^*$ to be fixed or change very slowly (adiabatically), that is, on the timescales longer than $\tau_s$. In that case, $J(t)$ and $h(t)$ trail the driving force $R^*(t)$, staying close to the corresponding slowly drifting fixed point $(J^*, h^*)$ in their 2D phase space:

$$h^* = \sqrt{R^*} - 1 \tag{S34}$$

$$J^* = \frac{k_0 h^* (1 + h^*)}{\tau_s} \tag{S35}$$

The stability of this slowly drifting fixed point and the more rapid epidemic dynamics can be described by linearizing *Equations S30* and *S32* around $(J^*, h^*)$, that is, by assuming $h(t) = h^* + \delta h(t)$ and $J(t) = J^* + \delta J(t)$:

$$\frac{d}{dt} \begin{pmatrix} \delta h \\ \delta J \end{pmatrix} = \frac{1}{\tau_s} \begin{pmatrix} -(1 + 2h^*) & \tau_s/k_0 \\ -2k_0 \gamma h^* & 0 \end{pmatrix} \begin{pmatrix} \delta h \\ \delta J \end{pmatrix} \tag{S36}$$

The eigenmodes of this linearized system are both stable, but the rates have substantial imaginary components:

$$r_\pm = -\frac{1 + 2h^*}{2\tau_s} \pm i \sqrt{\frac{2h^* \gamma}{\tau_s} - \frac{(1 + 2h^*)^2}{4\tau_s^2}} \tag{S37}$$

This indicates that relaxation towards point $(J^*, h^*)$ has a pronounced oscillatory character. The period of the oscillations is

$$T \approx \pi \sqrt{\frac{2\tau_s}{\gamma h^*}} \approx \pi \sqrt{\frac{2\tau_s}{\gamma \left( \sqrt{R^*} - 1 \right)}} \tag{S38}$$

The amplitude of the oscillations decays with the time constant $2\tau_s/(1 + 2h^*)$. This oscillatory behavior would manifest itself as multiple epidemic waves. In reality, the dynamics are more complicated since rapid changes of $M(t)$, for example, due to seasonal effects, government and societal response to the epidemic, would additionally modulate it.

The assumption of $R^* = R_0 M(t) S^\lambda(t)$ being fixed is not, of course, realistic. In particular, the mitigation factor $M(t)$ may have both slow and fast variations. On top of that, the dependence of $R^*$ on $S(t)$ creates a negative feedback suppressing the forcing on the long run. For a constant mitigation $M$, there is a line of fixed points $(J, S, h) = (0, S, 0)$, for any $S \leq S_{HI} = (R_0 M)^{-1/\lambda}$. Here $1 - S_{HI}$ represents the long-term HIT for a given mitigation level $M$. There is one particular solution $(\tilde{J}(t), \tilde{S}(t), \tilde{h}(t))$ corresponding to all three variables slowly evolving in such a way that $R_e$ stays close to 1 at all times, eventually reaching the HIT point, $(0, S_{HI})$. As follows from the above stability analysis, this solution acts as an attractor, with any trajectory in $(J, S, h)$ space converging towards it, unless perturbed by variations in mitigation $M(t)$. To construct that solution, we set the growth rate for $I(t)$ in *Equation S30* to 0. This gives

$$\frac{R_0 M \tilde{S}^\lambda}{(1 + \tilde{h})^2} = 1 \tag{S39}$$

By combining this result with *Equation S31*, one gets the following expression for $\tilde{J}(t)$:

$$\tilde{J}(t) = -\frac{1 + \tilde{h}}{\tilde{S}^{1/\kappa}} \cdot \frac{d \ln \tilde{S}}{dt} = -\frac{2}{\lambda} \left( \frac{R_0 M}{(1 + \tilde{h})^2} \right)^{1/(\lambda \kappa)} \frac{d\tilde{h}}{dt} \tag{S40}$$

After plugging it into *Equation S32* and taking the asymptotic limit $\tilde{h}(t) \ll 1$, one gets

$$\tau_s \left(1 + \frac{2(R_0 M)^{1/(\lambda\kappa)}}{\lambda k_0}\right) \frac{d\tilde{h}}{dt} = -\tilde{h}(t) \tag{S41}$$

This equation corresponds to the exponential decay of $\tilde{h}(t)$ with time constant given by

$$\tilde{\tau} = \tau_s \left(1 + \frac{2(R_0 M)^{1/(\lambda\kappa)}}{\lambda k_0}\right) \tag{S42}$$

Remarkably, under the assumption of strong overdispersion, $k_0 \ll 1$, the emergent timescale $\tilde{\tau}$ is significantly longer than the social rewiring time, $\tau_s$. This long timescale corresponds to a slow process of individuals trapped in the low-activity state, $a(t) \leq k_0$, transitioning to the high activity level $a \geq 1$. In the absence of persistent heterogeneity ($\kappa = \infty$, $\lambda = 1$), the new time constant is $\tilde{\tau} = \tau_s(1 + 2/k_0)$. For parameters of COVID-19 epidemic used in this study ($\kappa = 2$, $\lambda = 1.7$ and $R_0 M \simeq 2$), one gets $\tilde{\tau} = \tau_s(1 + 1.44/k_0)$, which for $k_0 = 0.4$ gives $\tilde{\tau} = 4.6\tau_s$.

The asymptotic long-term dynamics of $\tilde{h}(t)$, $\tilde{S}(t)$ and $\tilde{J}(t)$ are given by

$$\tilde{h}(t) = \tilde{h}_0 \exp(-t/\tilde{\tau}) \tag{S43}$$

$$\tilde{S}(t) = S_{HI} \left(1 + \frac{2}{\lambda}\tilde{h}(t)\right) \tag{S44}$$

$$\tilde{J}(t) \approx \left(\frac{1}{\tau_s} - \frac{1}{\tilde{\tau}}\right) k_0 \tilde{h}(t) \tag{S45}$$

where $\tilde{h}_0$ is a constant determined by the prior evolution of the epidemic.

Based on the stability analysis described in the previous section, for $M(t) = \text{const}$ any epidemic trajectory is bound to asymptotically converge to the universal attractor given by *Equations S43–S45*.

## Appendix 4

### Implementation of the agent-based model (ABM)

All simulations for the ABM use 1 million agents and three simulation replicates. For each agent in the simulation, at each time step, the social activity follows the stochastic dynamics described in *Equation 1*. After that, the overall force of infection is computed using

$$J(t) = \frac{\gamma R_0 M(t)}{\langle a(t)^2 \rangle_i} \frac{1}{N} \sum_i a_i I_i \qquad (S46)$$

where $I_i$ is binary and used to denote whether or not the agent is infectious, and $N$ is the number of agents in the simulation. For a susceptible agent , the chance of being infected in one simulation step is $a_i(t)J(t)dt$, which is proportional to the force of infection, his/her activity $a_i(t)$, and $dt$ – the length of the time step used in our simulations. For an infectious agent, the probability of recovering from the infectious state in one simulation step is $\gamma_0 dt$. Here, $\gamma_0$ is the transition rate between I to R compartments in the SIR model. When the waning of biological immunity is ignored, recovered agents will always stay in the recovered state and cannot be infected again.

## Appendix 5

### Waning of biological immunity

Our equations could be easily modified to account for the waning of biological immunity. This adds a new term in *Equation S31*, which becomes

$$\frac{dS}{dt} = -\frac{JS^{1+1/\kappa}}{(1+h)} + \frac{1-S}{\tau_b} \tag{S47}$$

Here, $\tau_b$ is the lifetime of biological immunity, which we set to 5 years throughout this work. The last term $\frac{1}{\tau_b}(1-S)$ describes the rate at which the recovered population (fraction $1-S$) reverts back to the susceptible state. The endemic steady state can be found by setting time derivatives *Equation S30*, *Equation S32* and *Equation S47*, to 0. Under the assumption that $\tau_b \gg \tilde{\tau}$, the endemic point in $(S, J, h)$ is given by

$$J_{en} \approx \frac{1 - S_{HI}}{\tau_b S_{HI}^{1+1/\kappa}} \tag{S48}$$

$$h_{en} \approx \tilde{\tau} J_{en} = \frac{\tilde{\tau}}{\tau_b} \frac{1 - S_{HI}}{S_{HI}^{1+1/\kappa}} \tag{S49}$$

$$S_{en} = S_{HI}(1 + h_{en})^{2/\lambda} \approx S_{HI} \tag{S50}$$

Here, $S_{HI} = R_0 M^{1/\lambda}$ corresponds to the HIT.

### Seasonal forcing

Seasonal effects are commonly described as a simple sin-shaped modulation of reproductive number (*Neher et al., 2020*). In this work, we used a combination of sigmoidal functions to model transition between 'winter' and 'summer' values of $M(t)$:

$$M_s(t) = 1 + \sigma \sum_{n=0}^{\infty} \left[ 1 - \tanh\left(\frac{t - t_{\text{spring}} + nT}{\Delta}\right) + \tanh\left(\frac{t - t_{\text{fall}} + nT}{\Delta}\right) \right] \tag{S51}$$

Here, $T = 1$ year, time parameters $t_{\text{spring}} < t_{\text{fall}}$ and $\Delta$ determine the timing of and sharpness of winter-summer-winter transitions. $\sigma$ determines the amplitude of seasonal changes. In particular, $\sigma = 0.25$ in *Figure 2*, and ranges between 0.25 and 0.35 in our fits of epidemic dynamics for different US regions (*Figure 7*).

### Mobility data for US regions

Historical mobility data for each of the 50 US states and the District of Columbia were downloaded from *China Data Lab Dataverse, 2021*. The data for four US regions were computed by weighting individual state data proportionally to their respective populations (*2020 US Census, 2021*).

### Fitting procedure for COVID-19 in the US regions

While fitting empirical data of daily COVID deaths in four US regions, we focused on the time from July 15, 2020, till February 25, 2021. The choice is motivated by Google Mobility Data that show only modest variation over that range, consistent with regular seasonal effects. Thus, function $R_0 M(t)$ has only three parameters: its summer and winter values, $R_{\text{summer}}$ and $R_{\text{winter}}$, as well as time of seasonal change $t_{\text{fall}}$, respectively. The width of transition was fixed at $\Delta = 30$ days.

Though outside of the range of interest, we have also fitted the epidemic curves for the earlier dates (March–July 2020). This was primarily done to ensure that the initial conditions $(J, S, h)$ in mid-July 2020 were consistent with the prior epidemic dynamics. As apparent from the mobility data, this early epidemic dynamics was strongly affected by government mitigation measures and collective knowledge-based response of the population. We were able to fit the overall daily death dynamics up to early July 2020 by varying initial incidence rate $j_0$ and three-parameter function $R_0 M(t)$. Specifically, the latter varied from $R_0 = 2.5$ to its lowest value $R_1$, relaxing later to post-mitigation level $R_2$. The original drop models both effects of government-imposed lockdowns and seasonal changes in spring 2020, and it is parameterized by transition time $t_{\text{spring}}$. Note that although formally $R_0 = 2.5$ is a fixed parameter, it is indirectly affected by varying $t_{\text{spring}}$. For the same reason, $t_{\text{spring}}$

should not be interpreted as an initial mitigation date as it depends on the choice of $R_0$. One should also be warned against overinterpreting behavior of $R_0M(t)$ at the early stages of the epidemics since *SIR* model becomes increasingly inadequate as $R_e$ significantly exceeds 1. On the other hand, this is not an issue for the entire date range of interest.

Thus, the overall epidemic curve in each US region has been fitted by our model with seven fitting parameters: four of them describing the early epidemic dynamics (March to early July 2020) and three parameterizing the seasonal changes within the date range of interest (July 2020–February 2021). The sets of fixed and best-fit model parameters are shown in *Appendix 5—table 1* and *Appendix 5—table 2*, respectively.

**Appendix 5—table 1.** The set of fixed model parameters used throughout this study.

| $k_0$ | $\kappa$ | $1/\gamma$ | $\tau_s$ | $\tau_b$ | **IFR** | $\Delta$ |
|-------|----------|-----------|----------|----------|---------|----------|
| 0.4 | 2 | 5 days | 30 days | 1 year | 0.5% | 30 days |

**Appendix 5—table 2.** The best-fit values for seven parameters in each of the four US regions. The fits were made using MATLAB R2021a nonlinear least-squares regression function (nlinfit)

| US region | $j_0$ | $R_1$ | $t_{spring}$ | $R_2$ | $R_{summer}$ | $R_{winter}$ | $t_{fall}$ |
|-----------|-------|-------|--------------|-------|--------------|--------------|------------|
| Northeast | 0.0014 | 1.14 | 26 Mar 2020 | 1.25 | 1.10 | 1.46 | 06 Oct 2020 |
| Midwest | 0.00036 | 0.95 | 20 Mar 2020 | 1.07 | 1.04 | 1.34 | 29 Sep 2020 |
| South | 0.00014 | 0.91 | 24 Mar 2020 | 1.43 | 1.04 | 1.33 | 07 Nov 2020 |
| West | 0.00015 | 0.97 | 15 Mar 2020 | 1.27 | 0.98 | 1.29 | 25 Oct 2020 |

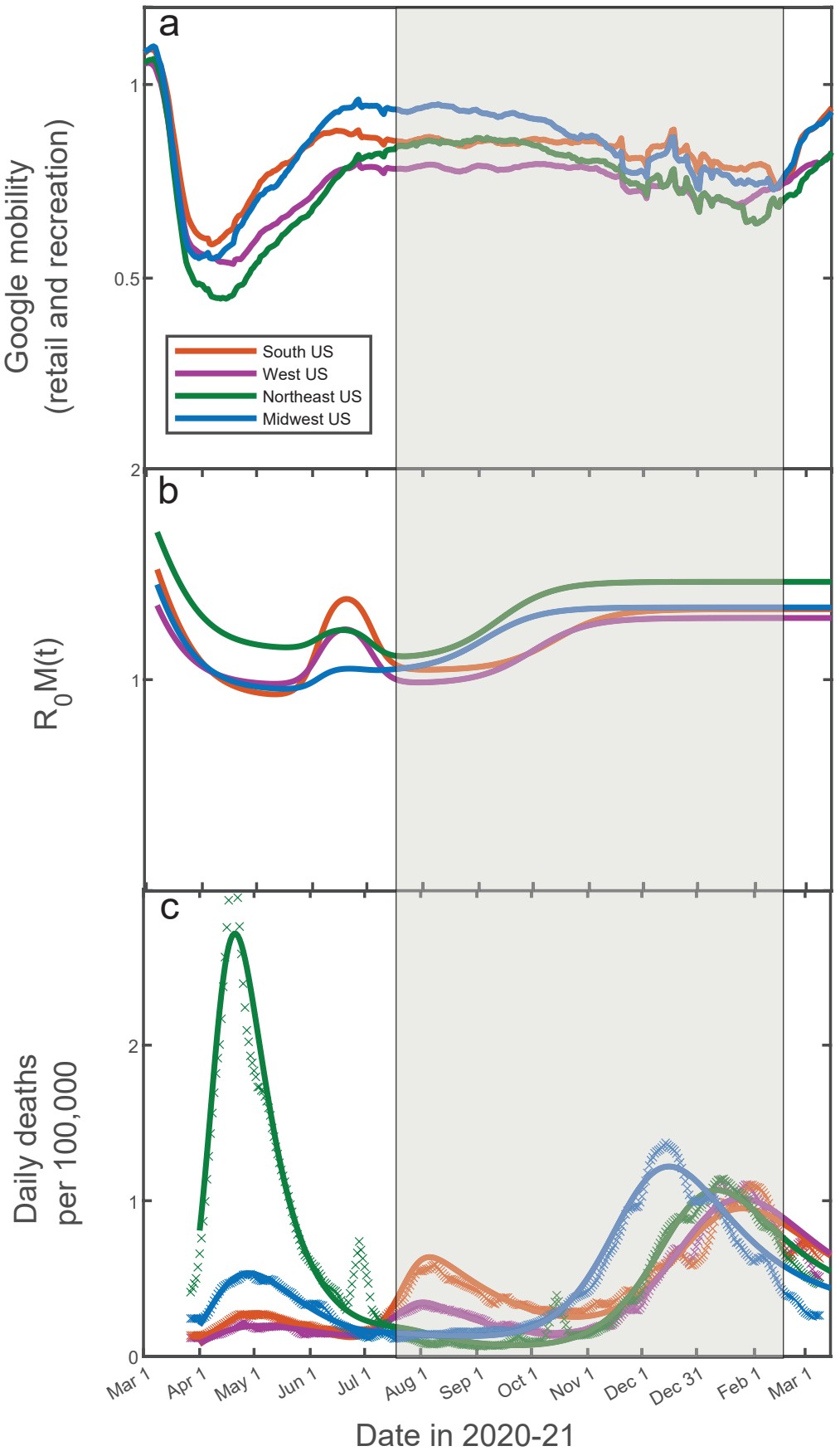

**Appendix 5—figure 1.** Fitting of the empirical data on COVID-19 epidemic in Northeast (green), Midwest (blue), West (purple), and South (orange) of the USA, alongside Google Mobility Data (**a**). Panel (**b**) shows the mitigation profile used in the model. Panel (**c**) shows the 7-day moving average of daily COVID-19 deaths fitted with the model. The time range of interest, presented in *Figure 3* of the main text, is shaded.

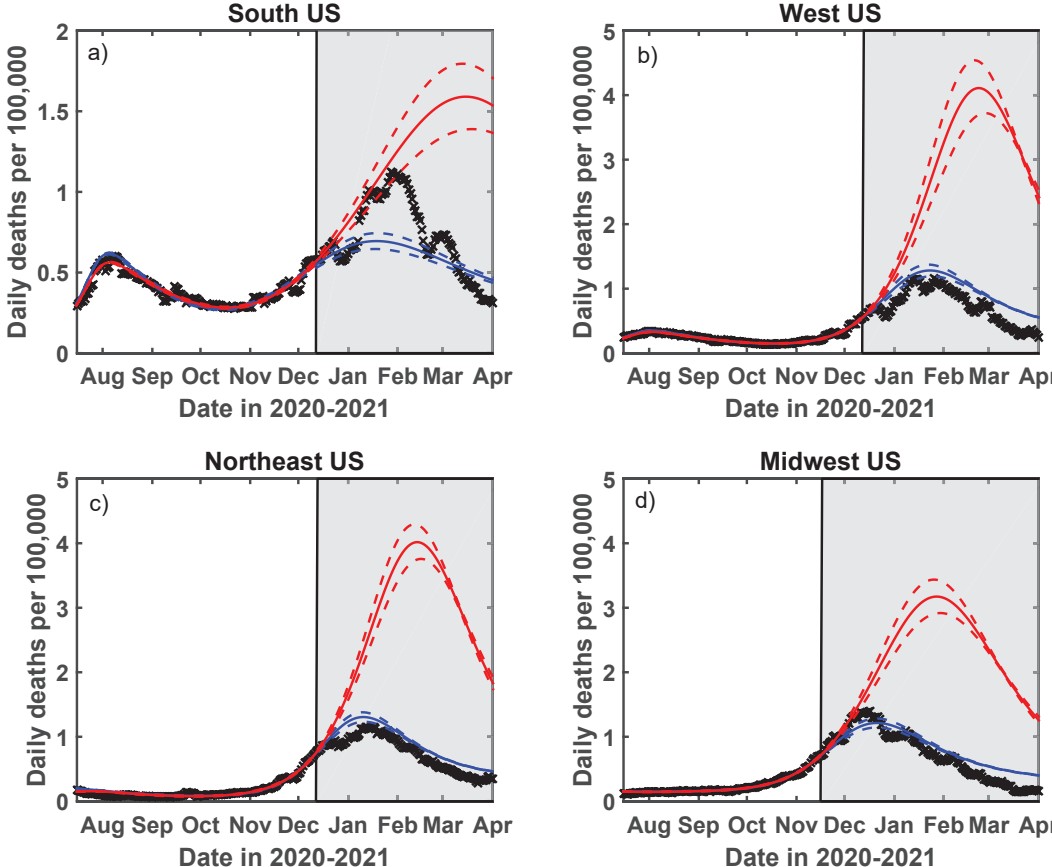

**Appendix 5—figure 2.** Test of the predictive power of the stochastic social activity (SSA) model developed in this work. Daily deaths data in each of four regions of the USA have been fitted up to December 13, 2020 (November 17, 2020, for the Midwest region). The epidemic dynamic beyond that date has been projected by our model (blue). One observes a good agreement between this prediction and the reported data (crosses). In contrast, the classical susceptible-infected-removed (SIR) model (red) substantially overestimates the height of the peak and projects it at a much later date than had been observed. Solid lines represent the best-fit behavior for each of the models, while dotted lines indicate the corresponding 95% confidence intervals.

## Sensitivity analysis with respect to $\tau_s$

We modified the nonlinear function used by the nlinfit MATLAB function to include $\tau_s$ among fitted parameters during the late epidemic dynamics. The set of best-fit values of $\tau_s$ in each of the four US regions along with the 95% confidence interval are shown in *Appendix 5—table 3*. The sensitivity analysis was carried out for $\tau_b = 5$ years.

**Appendix 5—table 3.** The best-fit values of $\tau_s$ in each of the four US regions.
The fits were made using MATLAB R2021a nonlinear least-squares regression function (nlinfit).

|  | Northeast | Midwest | South | West |
|---|---|---|---|---|
| Best fit (days) | 19 | 39 | 33 | 55 |

*Appendix 5—table 3 Continued on next page*

*Appendix 5—table 3 Continued*

|  | Northeast | Midwest | South | West |
|---|---|---|---|---|
| Lower 95% CI (days) | 17 | 36 | 27 | 49 |
| Upper 95% CI (days) | 20 | 42 | 38 | 61 |

The best-fit $\tau_s$ values in all of the regions range between 19 and 55 days. We verified that the overall agreement between the data and the model remains very good in each of the four regions for any choice of $\tau_s$ within that range (see *Appendix 5—figure 2*).

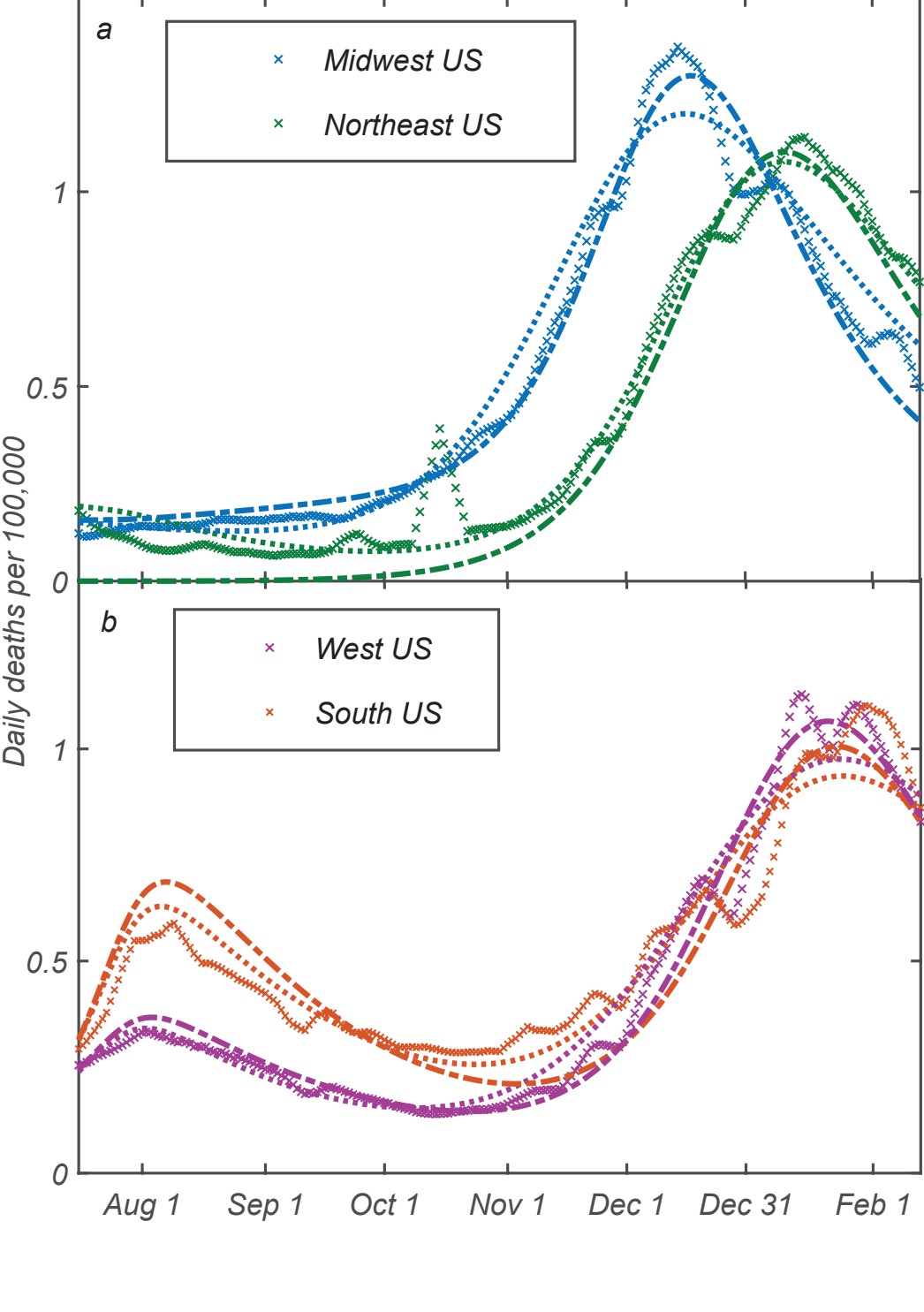

**Appendix 5—figure 3.** Analysis of the sensitivity of model predictions to parameter $\tau_s$. Dotted lines correspond to $\tau_s = 20$ days, while dot-dashed lines to $\tau_s = 55$ days. These values in turn correspond to the range of the best-fit values of $\tau_s$ in individual US regions (see **Appendix 5—table 3**).

