## [Editor Report]

This is an excellent and elegant example of what theory can do at its best in epidemiology: it takes a widely observed phenomenon that is an ‘embarrassment’ (my word) to current theories; proposes a parsimonious explanation that is plausible for the phenomenon by extending the existing theories in a specific way; and makes a plausible case for the importance of the mechanism in explaining key features of the data. In this case, the embarrassing phenomenon is long periods of very slowly changing incidence/prevalence, and the modification to theory is incorporation of dynamic social heterogeneity. This should stimulate much further work in the field. Congratulations to the authors.

---

## [Decision Letter]

**Decision letter after peer review:**

Thank you for submitting your article "Stochastic social behavior coupled to COVID-19 dynamics leads to waves, plateaus and an endemic state" for consideration by *eLife*. Your article has been reviewed by 2 peer reviewers, and the evaluation has been overseen by a Reviewing Editor and Aleksandra Walczak as the Senior Editor. The following individual involved in review of your submission has agreed to reveal their identity: Jennie Lavine (Reviewer #3).

Essential revisions:

1. From the editor: This seems to be a very important theoretical advance.

It is written like a physics paper, and will provide unnecessary (stylistic, terminologic) obstacles to infectious disease modelers trying to understand and build on it. Strongly encourage you to find someone in the field that will use this kind of work post-COVID and have them help you explain what you have done. Major examples:

– p. 6 is very hard going. What is an "immunity factor"? This is not a biology word. Do you really mean attack rate? That is not the same as the number of people infected per day, and the heterogeneity in it is just hard to understand. Please, for the sake of your citations and longevity of this paper, translate it into epidemiology with the help of someone who would be a reader in the more applied field.

2. Please clarify and strengthen the claims of explanatory power.

A principal concern about the paper is the implicit claim that the model explains the epidemiological patterns of COVID-19 in the United States during summer and fall 2020.

The authors fit their model to US death data by estimating parameters related to the degree of mitigation as a function of time M(t), as well as some seasonality parameters affecting R_0_ as a function of time. It is not clear whether baseline R_0_ was also estimated, since it is not listed as a fixed.

As the authors point out, monotonically increasing R_0_M(t) in a standard well-mixed SIR far from herd immunity would result in a single peak that overshoots the (ever-increasing) HIT. In the authors' fitted model, deaths in fact initially decline in the northeast and midwest before rising again, and the epidemic in the south displays two peaks separated by a trough.

But it is not clear this is a particularly convincing demonstration of the correctness of a model as an explanation for the observed dynamics. Official distancing policies may have monotonially become more lax over the period June 1 through to, e.g., the fall. But restrictions were tightened in winter in response to surges, and there was clear signal of behavioral response to increasing transmission that seems unlikely to have been mere regression to the mean.

In the model, the mitigation function is fitted; no actual data on deliberate versus randomly- varying behavior change is used. Given clear empirical signals of synchronous and deliberate response to epidemiology, modulated by social factors (Weill et al., 2020), a persuasive demonstration that consideration of random behavioral variation is necessary and/or sufficient to explain observed US COVID-19 dynamics would need to start from mobility data itself, and then find some principled way of partitioning changes in mobility into those attributable to random variation versus deliberate (whether top-down or bottom-up) action.

3. A further main concern is that the central result of transient epidemiological dynamics due to transient concordance of abnormally high versus low social activity-stems from the choice to model social behavior as stochastic but also mean-seeking. While is this idealization plausible, it would be valuable to motivate it more.

In other words, the central, compelling message of the paper is that if collective activity levels sometimes spike and crash, but ultimately regress to the mean, so will transmission. The more that behavioral model can be motivated, the more compelling the paper will be.

4. Line 48: It seems to me that the dynamic heterogeneity you incorporate does involve feedback from the current number of infections through the dependence of h(t) on J(t), which might act as a form of knowledge-based adaptation. Please explain this point and include a biological description of how you generated the h(t) term.

5. How sensitive are the qualitative results to different values of τ_s_?

6. Line 68: DIV(2020) – this citation is not in the references. Given that you also cite this for the data you plot, please include more details on where the data come from.

7. Line 222: The emergent long time constant seems to depend only on τ_s_ and k_0_ – is that correct? I would have thought the relaxation might also be affected by how rapidly the disease spread (i.e., M*J). This time scale is interesting and of relevance to public health measures, as it suggests when we might be reaching a sustainable plateau. Can you explain this in more detail?

*Reviewer #1 (Recommendations for the authors):*

1. Formal analysis and interpretation should better link the main text and the appendix In general, I would encourage the authors to link their formal model analysis in the Appendix more explicitly to the main text results. When a result is presented in the main text, the reader should be pointed to its derivation or justification in the appendix.

Similarly, to aid the less mathematical reader, it would be nice to interpret equations like S2 somewhat more when they are stated. For S2, for example, one could point out that the C(*τ*)*a_i_*(*t*) term reflects the individual's probability of infecting others conditional on having been infected τ units of time ago, while the *a_i_*(*t − τ*)*S_i_*(*t − τ*)*J*(*t − τ*) term reflects the probability that they were in fact infected τ units of time ago. With that in mind, I might adjust the notation slightly to highlight this by moving the *J*(*t − τ*) term into the average (though of course it can be factored out, as it does not depend on i) and grouping the two sets of terms in parentheses.

Moreover, having done the hard work of obtaining exact and/or approximate analytical results for their model, the authors should interpret these expressions more for the reader. e.g. the result about the HIT and λ in Equation 6 should be interpreted more in terms of the capacity for persistent heterogeneity to suppress the herd immunity threshold below the well-mixed case, and the contribution of even transient heterogeneity to determining the effective HIT.

2. Model definition. When introducing mathematical results and concepts in the main text, please make an explicit link to the corresponding Appendix derivations.

3. Code. In line with *eLife* guidelines, it would be good to provide the code used for model fitting, numerical solutions of differential equations, and stochastic simulations.

4. Undefined parameter γ Where is the parameter γ coming from in Equations S35 and subsequent? It is never defined. The other terms seem correct. Is this a holdover from a previous parametrization?

5. Model fitting. Model fitting procedures used to generate Figure 6 should be described in more detail in the appendix, and code should ideally be provided.

*Reviewer #3 (Recommendations for the authors):*

Line 48: It seems to me that the dynamic heterogeneity you incorporate does involve feedback from the current number of infections through the dependence of h(t) on J(t), which might act as a form of knowledge-based adaptation. Please explain this point and include a biological description of how you generated the h(t) term.

Line 68: DIV(2020) – this citation is not in the references. Given that you also cite this for the data you plot, please include more details on where the data come from.

How sensitive are the qualitative results to different values of τ_s_?

Line 222: The emergent long time constant seems to depend only on τ_s_ and k_0_ – is that correct? I would have thought the relaxation might also be affected by how rapidly the disease spread (i.e., M*J). This time scale is interesting and of relevance to public health measures, as it suggests when we might be reaching a sustainable plateau. Can you explain this in more detail?

---

## [Author Response]

Essential revisions:1. From the editor: This seems to be a very important theoretical advance.It is written like a physics paper, and will provide unnecessary (stylistic, terminologic) obstacles to infectious disease modelers trying to understand and build on it. Strongly encourage you to find someone in the field that will use this kind of work post-COVID and have them help you explain what you have done. Major examples:– p. 6 is very hard going. What is an "immunity factor"? This is not a biology word. Do you really mean attack rate? That is not the same as the number of people infected per day, and the heterogeneity in it is just hard to understand. Please, for the sake of your citations and longevity of this paper, translate it into epidemiology with the help of someone who would be a reader in the more applied field.

We now discuss the immunity factor together with other results from our previous paper (PNAS 2021) in the introduction section. All major concepts are defined so that readers don't need to refer to that earlier work. Also, to streamline the discussion on page 6 we moved the complex equation for the immunity factor to the Appendix.

We were able to get feedback on our manuscript from Neil Ferguson (Imperial College London) and addressed his comments in the revised version.

2. Please clarify and strengthen the claims of explanatory power.A principal concern about the paper is the implicit claim that the model explains the epidemiological patterns of COVID-19 in the United States during summer and fall 2020.The authors fit their model to US death data by estimating parameters related to the degree of mitigation as a function of time M(t), as well as some seasonality parameters affecting R_0_ as a function of time. It is not clear whether baseline R_0_ was also estimated, since it is not listed as a fixed.

The epidemic dynamics is determined by the combination R_0_*M(t). The unmitigated R_0_ reveals itself only during the very early phases of the epidemics before any social distancing was in place. Therefore, within the time interval considered in our study we cannot separately determine R_0_ and M(t).

As the authors point out, monotonically increasing R_0_M(t) in a standard well-mixed SIR far from herd immunity would result in a single peak that overshoots the (ever-increasing) HIT. In the authors' fitted model, deaths in fact initially decline in the northeast and midwest before rising again, and the epidemic in the south displays two peaks separated by a trough.But it is not clear this is a particularly convincing demonstration of the correctness of a model as an explanation for the observed dynamics. Official distancing policies may have monotonially become more lax over the period June 1 through to, e.g., the fall. But restrictions were tightened in winter in response to surges, and there was clear signal of behavioral response to increasing transmission that seems unlikely to have been mere regression to the mean.In the model, the mitigation function is fitted; no actual data on deliberate versus randomly- varying behavior change is used. Given clear empirical signals of synchronous and deliberate response to epidemiology, modulated by social factors (Weill et al., 2020), a persuasive demonstration that consideration of random behavioral variation is necessary and/or sufficient to explain observed US COVID-19 dynamics would need to start from mobility data itself, and then find some principled way of partitioning changes in mobility into those attributable to random variation versus deliberate (whether top-down or bottom-up) action.

We agree with the referee that multiple factors affect the epidemic dynamic. These include: government imposed mitigation, knowledge-based adaptation of social behavior, seasonal forces, vaccination, emergence of new variants, etc. Constructing and calibrating a model taking into account all of these factors is well beyond the scope of this study focused on stochastic changes in social activity. As pointed out by the referee, a principled way of integrating effects of mitigation and knowledge-based adaptation is to use the average mobility data available from Google or other providers. By their nature, these data capture population-wide trends in social activity, while averaging out individual level stochasticity. In this work, we take advantage of the observation that the average mobility across the US (see the new Figure 6) remained remarkably steady during the period considered in our study (July 2020- February 2021). Hence, this time interval is optimal for testing the predictions of our theory without embarking on calibration of a full scale multi-parameters model.

Another major addition to our analysis in the revised version of the manuscript is the new Figure 8 (as well as Appendix 5. Figure 2) comparing the predictions of our model with the classical SIR model. As one can see from this figure, our model is capable of predicting the shape of the epidemic wave based on the data covering its early stages. This should be contrasted with the predictions of the SIR model dramatically overshooting the observed maximum of daily deaths. This is consistent with the Transient Collective Immunity proposed and analyzed in this study as well as our earlier publication (Tkachenko et al. PNAS (2021)).

3. A further main concern is that the central result of transient epidemiological dynamics due to transient concordance of abnormally high versus low social activity-stems from the choice to model social behavior as stochastic but also mean-seeking. While is this idealization plausible, it would be valuable to motivate it more.In other words, the central, compelling message of the paper is that if collective activity levels sometimes spike and crash, but ultimately regress to the mean, so will transmission. The more that behavioral model can be motivated, the more compelling the paper will be.

We included an additional justification of our form of stochastic social dynamics and expanded the discussion of relevant prior studies. Especially revealing are the studies of burstiness in virtual communication such as email (Vazquez et al. (2007); Karsai et al. (2012)). Studies of digital communications can be easily studied over a substantial time interval, which is more problematic for field studies of face-to-face contact networks. These studies unequivocally show the regression of individual activity levels towards its long-term mean value. This regression happens over a well-defined relaxation time ranging from days to months depending on the context. Note that the value towards which the activity regresses may not be identical for different individuals. In the context of our model, such persistent heterogeneity is captured by the distribution of α_i_ with the dispersion parameter κ.

4. Line 48: It seems to me that the dynamic heterogeneity you incorporate does involve feedback from the current number of infections through the dependence of h(t) on J(t), which might act as a form of knowledge-based adaptation. Please explain this point and include a biological description of how you generated the h(t) term.

As now explained in the text (see the caption to Figure 2), this feedback mechanism is due to the selective removal of susceptibles with high current levels of social activity in the course of the epidemic. Therefore, it does not involve any knowledge-based adaptation, defined as modulation of average social activity in response to the perceived danger of the current level of infection.

5. How sensitive are the qualitative results to different values of τ_s_?

We conducted the sensitivity analysis with respect to τ_s_ and presented the results of this analysis in the main text as well as in a new section in Appendix 5 (see Appendix 5. Figure 2).

6. Line 68: DIV(2020) – this citation is not in the references. Given that you also cite this for the data you plot, please include more details on where the data come from.

Done.

7. Line 222: The emergent long time constant seems to depend only on τ_s_ and k_0_ – is that correct? I would have thought the relaxation might also be affected by how rapidly the disease spread (i.e., M*J). This time scale is interesting and of relevance to public health measures, as it suggests when we might be reaching a sustainable plateau. Can you explain this in more detail?

We have substantially expanded the explanation of the origin of this emergent timescale and its implications for public policy measures. Furthermore, we have better explained its derivation in Appendix 3 (see the red text in the newly formed section "Long-term plateau dynamics"). The editor is correct in pointing out that this new timescale (τ~) is affected by the basic reproduction number corrected by the mitigation: M*R_0_. This dependence is explicitly presented in Appendix 3 (see Equation S42). However, this dependence is relatively weak. For COVID-19, we estimated this timescale to be approximately 5* τ_s_. Also, there is no dependence of this timescale on the *current* strength of the epidemic M*J, since the latter is itself exponentially decaying with the same time constant (unless waning of biological immunity is taken into account).

Reviewer #1 (Recommendations for the authors):1. Formal analysis and interpretation should better link the main text and the appendix In general, I would encourage the authors to link their formal model analysis in the Appendix more explicitly to the main text results. When a result is presented in the main text, the reader should be pointed to its derivation or justification in the appendix.Similarly, to aid the less mathematical reader, it would be nice to interpret Equations like S2 somewhat more when they are stated. For S2, for example, one could point out that the C(τ)a_i_(t) term reflects the individual's probability of infecting others conditional on having been infected τ units of time ago, while the a_i_(t − τ)S_i_(t − τ)J(t − τ) term reflects the probability that they were in fact infected τ units of time ago. With that in mind, I might adjust the notation slightly to highlight this by moving the J(t − τ) term into the average (though of course it can be factored out, as it does not depend on i) and grouping the two sets of terms in parentheses.

We corrected the text as suggested and added explanations of individual terms in Equation S2.

Moreover, having done the hard work of obtaining exact and/or approximate analytical results for their model, the authors should interpret these expressions more for the reader. e.g. the result about the HIT and λ in Equation 6 should be interpreted more in terms of the capacity for persistent heterogeneity to suppress the herd immunity threshold below the well-mixed case, and the contribution of even transient heterogeneity to determining the effective HIT.

In response to suggestion of other referees and our editor we decided to remove the Equation 6 from the main text. In addition we expanded the discussion of the results from our earlier publication (PNAS 2021) in the introduction. We specifically focused on λ and its impact on HIT.

2. Model definition. When introducing mathematical results and concepts in the main text, please make an explicit link to the corresponding Appendix derivations.

We completely restructured the Appendix by breaking it down into several independent "boxes" with titles aligned to the main text of the article. Furthermore, these separate Appendix boxes are now properly referenced in the main text greatly facilitating the search of the corresponding mathematical derivations.

3. Code. In line with eLife guidelines, it would be good to provide the code used for model fitting, numerical solutions of differential equations, and stochastic simulations.

Done. Our code for the Agent Based Model and the US regions fits were added to the public Github repository: https://github.com/maslov-group/COVID-19-waves-and-plateaus/.

4. Undefined parameter γ Where is the parameter γ coming from in Equations S35 and subsequent? It is never defined. The other terms seem correct. Is this a holdover from a previous parametrization?

The parameter γ was indeed a holdover from a previous parameterization. It has now been replaced with 1/τ_g._

5. Model fitting. Model fitting procedures used to generate Figure 6 should be described in more detail in the appendix, and code should ideally be provided.

Our fitting procedures are now explained in Appendix 5 and the Matlab code for generating the fits is available on Github.

Reviewer #3 (Recommendations for the authors):Line 48: It seems to me that the dynamic heterogeneity you incorporate does involve feedback from the current number of infections through the dependence of h(t) on J(t), which might act as a form of knowledge-based adaptation. Please explain this point and include a biological description of how you generated the h(t) term.

As now explained in the text (see the caption to Figure 2), this feedback mechanism is due to the selective removal of susceptibles with high current levels of social activity in the course of the epidemic. Therefore, it does not involve any knowledge-based adaptation defined as modulation of average social activity in response to the perceived danger of the current level of infection.

Line 68: DIV(2020) – this citation is not in the references. Given that you also cite this for the data you plot, please include more details on where the data come from.

Corrected.

How sensitive are the qualitative results to different values of τ_s_?

We conducted the sensitivity analysis by varying τ_s_ from 15 days to 90 days and presented the results of this analysis in a new supplementary figure.

Line 222: The emergent long time constant seems to depend only on τ_s_ and k_0_ – is that correct? I would have thought the relaxation might also be affected by how rapidly the disease spread (i.e., M*J). This time scale is interesting and of relevance to public health measures, as it suggests when we might be reaching a sustainable plateau. Can you explain this in more detail?

We have substantially expanded the explanation of the origin of this emergent timescale and its implications for public policy measures. Furthermore, we have better explained its derivation in the Appendix 3 (see the red text in the newly formed section "Long-term plateau dynamics"). The editor is correct in pointing out that this new timescale (tilde tau) is affected by the basic reproduction number corrected by the mitigation: M*R_0_. This dependence is explicitly presented in the Appendix 3 (see Equation S42). However, this dependence is relatively weak. For COVID-19 we estimated this timescale to be approximately 5* τ_s_. Also, there is no dependence of this timescale on the *current* strength of the epidemic M*J, since the latter is itself exponentially decaying with the same time constant (unless waning of biological immunity is taken into account).